# Dissecting the role of bHLH transcription factors in the potato spindle tuber viroid (PSTVd)-tomato pathosystem using network approaches

Katia Aviña-Padilla[1,2], Octavio Zambada-Moreno[1], Marco A. Jimenez-Limas[3], Rosemarie W. Hammond[4], Maribel Hernández-Rosales[1]*

**1** Deparment of Genetic Engineering, Center for Research and Advanced Studies (Cinvestav), Irapuato, Guanajuato, Mexico, **2** Department of Crop Sciences, University of Illinois at Urbana–Champaign, Urbana, Illinois, United States of America, **3** Center for Research in Computation, National Polythecnic Institute, Mexico City, Mexico, **4** United States of America Department of Agriculture, Beltsville Agricultural Research Center, Beltsville, Maryland, United States of America

* maribel.hr@cinvestav.mx

## Abstract

Viroids, minimalist plant pathogens, pose significant threats to crops by causing severe diseases. Transcriptome profiling technologies have significantly advanced the analysis of viroid-infected host plants, providing critical insights into gene regulation by these pathogens. Despite these advancements, the presence of numerous genes of unknown function continues to limit a complete understanding of the transcriptome data. Co-expression analysis addresses this issue by clustering genes into modules based on global gene expression levels, with genes in the same cluster likely participating in the same biological pathways. In a previous study, we emphasized the importance of basic helix-loop-helix (bHLH) proteins in transcriptional reprogramming in tomato host in response to different potato spindle tuber viroid (PSTVd) strains. In the current research, we delve into tissue-specific gene modules, particularly in root and leaf tissues, governed by bHLH transcription factors (TFs) during PSTVd infections. Utilizing public datasets that span Control (C), mock-inoculated, PSTVd-mild (M), and PSTVd-severe (S23) strains in time-course infections, we uncovered differentially expressed gene modules. These modules were functionally characterized to identify essential hub genes, notably highlighting the regulatory coordination of bHLH TFs, depicted through the significant bifan motif found in these interactions. Expanding on these findings, we explored bipartite networks, discerning both common and unique bHLH TF regulatory roles. Our findings reveal that bHLH TFs play pivotal roles in regulating processes such as energy metabolism and facilitating rapid membrane repair in infected roots. In leaves, changes in the external layers affected photosynthesis, linking bHLH TFs to distinct metabolic functions. Through this holistic approach, we deepen our understanding of viroid-host interactions and the intricate regulatory mechanisms underpinning them.

**Data availability statement:** All data supporting the findings of this study are available in the repository: https://github.com/kap8416/Dynamic-coexpression-modular-analysis-BHLH-in-PSTVD-tomato

**Funding:** This research was funded by internal USDA-ARS project number 8042-22000-318-00D. K.A.-P. (CVU:227919), O.Z.-M. (CVU:1147042) and M.A.J.-L. (CVU:1035685) received financial support from the CONAHCyT. K.A.-P. had a fellowship from the Fulbright García-Robles foundation.

**Competing interests:** The authors have declared that no competing interests exist.

## 1. Introduction

Viroids are small, infectious RNAs that cause economically important diseases in plants [1,2]. Despite their tiny size of 246–434 nucleotides, they have a distinctive single-stranded circular RNA genome that operates devoid of coding capacity [3]. This lack of protein expression makes viroids unique among plant pathogens, relying entirely on host enzymatic machinery for replication and systemic movement. Operating independently, these minuscule agents hijack the host's molecular machinery to replicate and propagate within plant tissues [4]. Beyond their impact on crops, viroids serve as a valuable model for unraveling intricate host-pathogen interactions[5–10]. Moreover, viroid infections have significant repercussions as they disrupt host development and interfere with crucial physiological processes [11–15]. An illustrative instance can be observed in tomato (*Solanum lycopersicum*), a vital global agricultural commodity that yielded more than 130,812,947 million dollars in 2022 (http://faostat.fao.org/, accessed in May 2024). This staple crop faces a tangible threat from viroids, underscoring the potential impact of these infectious agents on essential food production systems. Tomato was selected as the model system for this study due to its economic importance as a staple crop worldwide and its well-characterized genetic and transcriptomic resources (https://www.solgenomics.net/). This choice is particularly relevant given that Potato spindle tuber viroid (PSTVd), one of the most studied viroids, has a significant economic impact on solanaceous crops, including tomato. PSTVd has been reported in major tomato-growing regions worldwide, including Europe, North and South America, Asia, and Africa, where it poses a significant threat to agricultural productivity (https://gd.eppo.int/taxon/PSTVD0/distribution, accessed on January 2025). Outbreaks in these regions are often associated with the international trade of infected plant material and inadequate phytosanitary controls, highlighting the importance of global surveillance and stringent management practices.

Beyond its agricultural impact, PSTVd exemplifies how viroid infections can profoundly alter host physiology. Emerging reports link viroid diseases hormone pathways and transcription factor dynamics, disrupting plant gene expression landscapes[16–18]. Transcriptional profiling has been instrumental in understanding the global effects of viroid infections. These high-throughput approaches have significantly advanced our understanding of the global effects of viroid infections, uncovering extensive alterations in gene expression across various plant systems [19]. Constructing co-expression networks across diverse conditions, such as disease states, facilitates the identification of disease-induced changes through network connectivity patterns [20–25]. This analysis groups genes into modules based on similar expression patterns, allowing the inference of functions for unknown genes clustered with well-characterized ones[25–31]. By examining these patterns, potential functions for unannotated genes, such as involvement in stress response, can be predicted. Network centrality measures help identify *"hub"* genes, which may indicate key regulatory roles [32–35]. Additionally, overlaying clusters in co-expression networks with pathway data can suggest unknown genes involvement in specific

biological pathways, providing further functional insights. Notably, genes with shared functions often exhibit robust correlations in their expression levels, laying the foundation for uncovering molecular pathways underlying diseases and conditions[33,34].

Genetic and molecular studies centered on plant TFs have provided invaluable insights into plant-specific responses, encompassing defense against pathogens, responses to light, and environmental stresses like cold, drought, high salinity, and developmental processes [36–38]. Transcriptional regulation, a fundamental process governing tissue-specific gene expression and responses to stimuli, underlies these processes [37]. bHLH TFs, prevalent in eukaryotes, constitute a substantial TF family. Anchored by a DNA-binding bHLH domain, these proteins orchestrate various biological processes, including stress responses[36–39]. The bHLH domain's N-terminus contains a DNA-binding motif, with proteins binding to sequences harboring the E-box (59-CANNTG-39) consensus motif, commonly the G-box (59-CACGTG-39).

In a previous study, we conducted a comprehensive analysis on the regulatory mechanisms in tomato plants infected by PSTVd, which highlighted the significant roles of bHLH TFs during infection [18]. Based on these insights, we decided to focus exclusively on these TFs due to their central role in regulating plant metabolism and defense. Our findings showed that bHLH-regulated modules were strain-specific, indicating unique regulatory adaptations to different PSTVd infections. Hence, we have selected bHLH TFs to further investigate guided regulatory transcriptional events.

In this study, we aimed to identify co-expression modules of the bHLH transcription factor family, with a particular emphasis on their roles in diverse biological processes, including the tomato immune response to viroid infection. By positioning bHLH TFs as hub genes, we explored their involvement in key host molecular mechanisms such as stress responses, development, and signaling pathways. Our primary objective was to uncover tissue- and strain-specific bHLH-guided transcriptional programs, highlighting both conserved and unique regulatory network mechanisms using network-based approaches.

Our results reveal that bHLH TFs are associated with regulation of genes involved in energy metabolism and membrane lipid repair functions in infected roots. Furthermore, our findings shed light on the intricate regulatory networks orchestrated by bHLH TFs in leaf tissue during PSTVd variant infections. The disruptions in cuticle function, shifts in metabolic processes, and alterations in biosynthesis pathways portray the plant's adaptive response to mitigate viroid impact. Notably, our findings emphasize the enrichment of the bifan motif, a critical regulatory interaction, conserved and intricately woven across the analyzed biological networks. The coordinated regulation of bHLHs through this motif ensures synchronized transcriptional responses, optimizing plant defense mechanisms under stress conditions. This comprehensive understanding contributes to unraveling the host-pathogen interaction and may provide strategies to enhance plant resilience against viroid infections.

## 2. Materials and methods

### 2.1. Designed pipeline for the networks approaches

We performed a comprehensive comparison of the effects exerted by PSTVd M (mild) and S23 (severe) strains on the susceptibility of the tomato host, with a primary focus on symptom development. To achieve this, we employed an approach that integrates transcriptomics and functional genomics data. Through this strategy, we implemented co-expression and module network analysis for root and leaf tissues, as depicted in **Fig 1**. Our designed codes are available at https://github.com/kap8416/Dynamic-coexpression-modular-analysis-BHLH-in-PSTVD-tomato. For the network analyses, we used Corto, CEMiTool, and MFinder tools due to their specialized capabilities. The Corto algorithm was used to reconstruct tissue-specific transcriptional regulatory networks, emphasizing bHLH interactions [40](Steps (A)–(B)), leveraging prior biological knowledge for accurate inference. Then, CEMiTool identified and enriched co-expression modules, providing system-wide insights into gene interactions [21](Step (C)). Bipartite networks for each tissue were then constructed using Python scripts derived from CEMiTool output files (Step (D)). Finally, MFinder detected network

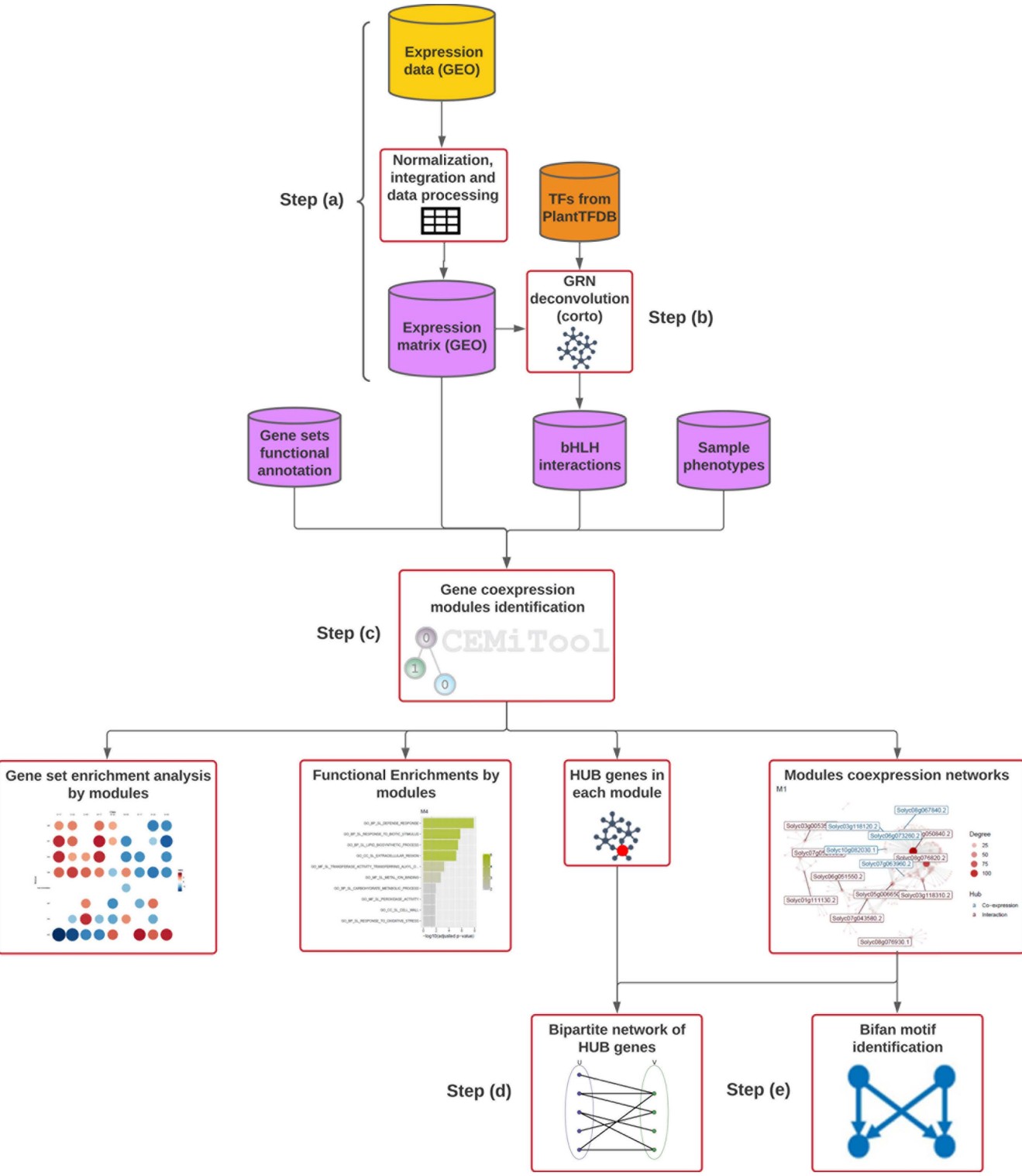

**Fig 1. Pipeline for the network approaches. (A)** We acquired publicly available microarray expression data encompassing both control and infected leaves and roots from the NCBI. **(B)** With the obtained expression data, we constructed an expression matrix of each tissue (roots, and leaf separately) and curated additional pertinent files, collectively serving as the foundational input for our subsequent analysis as follows: functional annotation, expression matrix, interest genes interactions, and sample phenotypes (the required input files are depicted in pink). **(C)** Then, we performed a co-expression modular network analysis using CEMiTool, a specialized tool for inferring co-expression modules. **(D)** Following the completion of the network analysis,

we delved into Gene Set Enrichment Analysis (GSEA) performed by CEMiTool to unveil significant associations, and functional enrichment analysis was executed on the modules. Furthermore, it identifies hub genes residing within the network clusters. **(E)** Additionally, we constructed bipartite networks delineating the bHLH-regulon interactions. **(G)** Finally, we identified the most representative motifs within each network module.

motifs within the co-expression networks, uncovering key regulatory patterns [41] (Step (E)). These tools were chosen for their precision, ease of implementation, and alignment with the study's objectives. Time points (17, 24, and 49 dpi) were analyzed differently depending on the network type. In the global GRN, data from all-time points were integrated to create a comprehensive co-expression network. For module-specific networks, time points were treated separately to capture dynamic changes across stages, resulting in dynamic networks. Finally, in the bipartite networks, data from all-time points were combined again to provide an integrated overview of significant interactions.

## 2.2. Description of the gene expression datasets

Two expression matrices depicting the comprehensive effects of PSTVd infection on the tomato plants' transcriptome were utilized. Microarray datasets from the roots (GSE111736) and leaves (GSE106912) of tomato plants under control (C), mild infection (M), and severe infection (S) conditions were obtained from the NCBI Gene Expression Omnibus database (https://www.ncbi.nlm.nih.gov/gds, accessed on June 4, 2022). These studies employed a time-course analysis spanning three distinct stages: early symptoms (17 days post-inoculation = dpi), pronounced symptoms (24 dpi), and recovery (49 dpi), resulting in a total of 26 root and 27 leaf samples being individually assessed [42,43]. Despite the advantages of high-throughput data, we selected these datasets because they represent the most extensive resource currently available for studying gene expression in tomato under viroid infection. Furthermore, the sample sizes for each dataset are sufficiently large to generate reliable coexpression networks, ensuring the identification of key regulatory nodes and gene modules involved in tissue-specific responses.

## 2.3. Obtaining the bHLH- gene specific interactions

The Corto algorithm, available on the CRAN repository of R packages, was used to infer gene regulatory networks (GRNs) from tissue-specific expression matrices. For this analysis, we used an integrated expression matrix and the PlantTFDB tomato transcription factor list. Corto employs Spearman correlation to identify gene associations and applies the Data Processing Inequality (DPI) to eliminate indirect interactions, with bootstrapping to ensure robustness. The leaf network included 27 samples and 8080 features, while the root network had 26 samples and 8080 features. Spearman correlation thresholds of 0.837 (leaves) and 0.828 (roots) were applied, alongside a *p-value* threshold of $1 \times 10^{-8}$ was applied to filter significant edges, and 100 bootstraps were performed to ensure robustness. From these global GRNs, subnetworks focusing on bHLH-TFs and their interactors were extracted, enabling detailed analysis of tissue-specific regulatory dynamics under PSTVd-induced stress

## 2.4. Modular co-expression network analysis

For the identification of gene co-expression modules, we utilized the CEMiTool interface [https://cemitool.sysbio.tools accessed in May 2023]. The modular analysis was conducted by inputting the following datasets (**Fig 1C**):

1. A tab-delimited text gene expression matrix customized for each tissue.

2. A tab-delimited text table containing interaction details of the bHLH-TFs (bHLH GRN obtained in 2.3).

3. Tomato gene ontologies presented in GMT file format.

4. A text file table with "SampleName" and "Class" columns, providing descriptive information about the phenotype of each sample.

The analysis was executed with default parameter settings. Prior to analysis, the expression matrices underwent normalization using the RMA function from the limma R package [44]. This normalization aimed to derive significant Beta-values crucial for accurately inferring co-expression modules.

## 2.5. Identification of temporally differential modules

We first identified modules that showed contrasting expression patterns at different time points (dpi). For instance, modules that were repressed at 17 dpi but showed overexpression at 24 dpi were flagged for further analysis. This temporal shift in gene expression was a key indicator of modules responding dynamically to the infection process. We also compared the behavior of modules under infection by different strains of PSTVd. For example: modules that exhibited a different expression behavior based on the normalization enrichment score (NES) value (e.g., activation or repression) in response to PSTVd-M, and an opposite or contrasting behavior in response to PSTVd-S, were selected.

## 2.6. Obtaining bHLH bipartite networks and functional assignment

Based on the outcomes of the modular co-expression analysis, we constructed bipartite networks for the different plant tissues. Each network comprises two integral layers, the first layer includes all bHLH TFs acting as regulators, and the second layer encompasses their respective target genes (regulon). This method allows for a detailed visualization of how these bHLH factors influence gene expression across tissues. These networks were created utilizing Python scripts available at https://github.com/kap8416/Dynamic-coexpression-modular-analysis-BHLH-in-PSTVD-tomato. For the visualizations, all the bHLH-TFs involved in each tissue were represented, alongside the genes they regulate, referred to as the regulon. These components were categorized based on the way their regulation occurs. Moreover, we pursued functional enrichment analysis by employing gene ontology terms for distinct categories of regulation. To execute this step, we employed the Gprofiler package [45] within the RStudio environment using an FDR threshold of < 0.05 to ensure statistically significant results. This enabled us to discern and characterize the functional attributes associated with the regulatory dynamics observed within the bipartite networks.

## 2.7. Identifying regulatory motifs within network modules

After obtaining the network modules for both leaf and root tissues, we searched for significant regulatory motifs within each of these modules. To achieve this, we utilized the Mfinder tool, a specialized resource designed for detecting network motifs. This tool employs two distinct methods for motif identification: 1) Full enumeration of subgraphs: this technique involves an exhaustive enumeration of subgraphs within the network. The concept of network motifs, seen as fundamental building blocks of complex networks, forms the basis of this approach [46]; 2) Sampling of subgraphs for concentration estimation: in this method, subgraphs are sampled to estimate their concentrations within the network. By assessing prevalence and significance, this approach aids in detecting network motifs [41]. To execute this process, a.txt file containing the module of interest was submitted to the Mfinder tool. The tool's output comprises significant motifs identified within the network.

# 3. Results

## 3.1. Bipartite networks: unveiling specific regulatory patterns

Our study employs a global analysis strategy utilizing bipartite networks to dissect the intricate interactions of bHLH TFs in plant response mechanisms. Specifically, the leaf tissue bipartite network comprises 30 bHLH genes, while the root tissue network contains 28. A significant aspect of our analysis is the overlap of 28 bHLH genes between these two networks, indicating a core regulatory mechanism common to both tissues. However, the leaf network extends beyond this commonality by including two additional bHLH genes *Solyc03g097820.1* SlbHLH022 and *Solyc06g072520.1* GBOF-1, which are

linked to key physiological, developmental, and metabolic processes; and drought response, respectively. The observed differential composition between the leaf and root networks suggests that bHLH TFs may have tissue-specific regulatory roles, potentially influencing distinct physiological, developmental, and metabolic processes in a tissue-dependent manner. However, further experimental validation is required to confirm these roles and their underlying mechanisms.

**3.1.1. Insights from the root bipartite network.** For root samples, the bipartite network comprises 28 bHLHs and 1004 target genes. This network reveals 563 genes regulated by one bHLH, 316 targets regulated by two bHLHs, 93 targets regulated by three bHLHs, and 32 targets regulated by four bHLHs—none of the genes are regulated by more than four bHLHs. Functional enrichment analysis sheds light on the roles of these regulons. Genes with a single regulator are functionally enriched in *Chloroplast, Photosystems I and II* functions, while genes regulated by two bHLHs play a part in *KEGG 00592 Alpha-linoleic acid metabolism*, as shown in **Fig 2**. The finding that genes with a single regulator are enriched in chloroplast, *Photosystems I and II functions* suggests a specialized regulatory mechanism. Even though chloroplasts are predominantly found in photosynthetic tissues like leaves, these genes may still play a role in

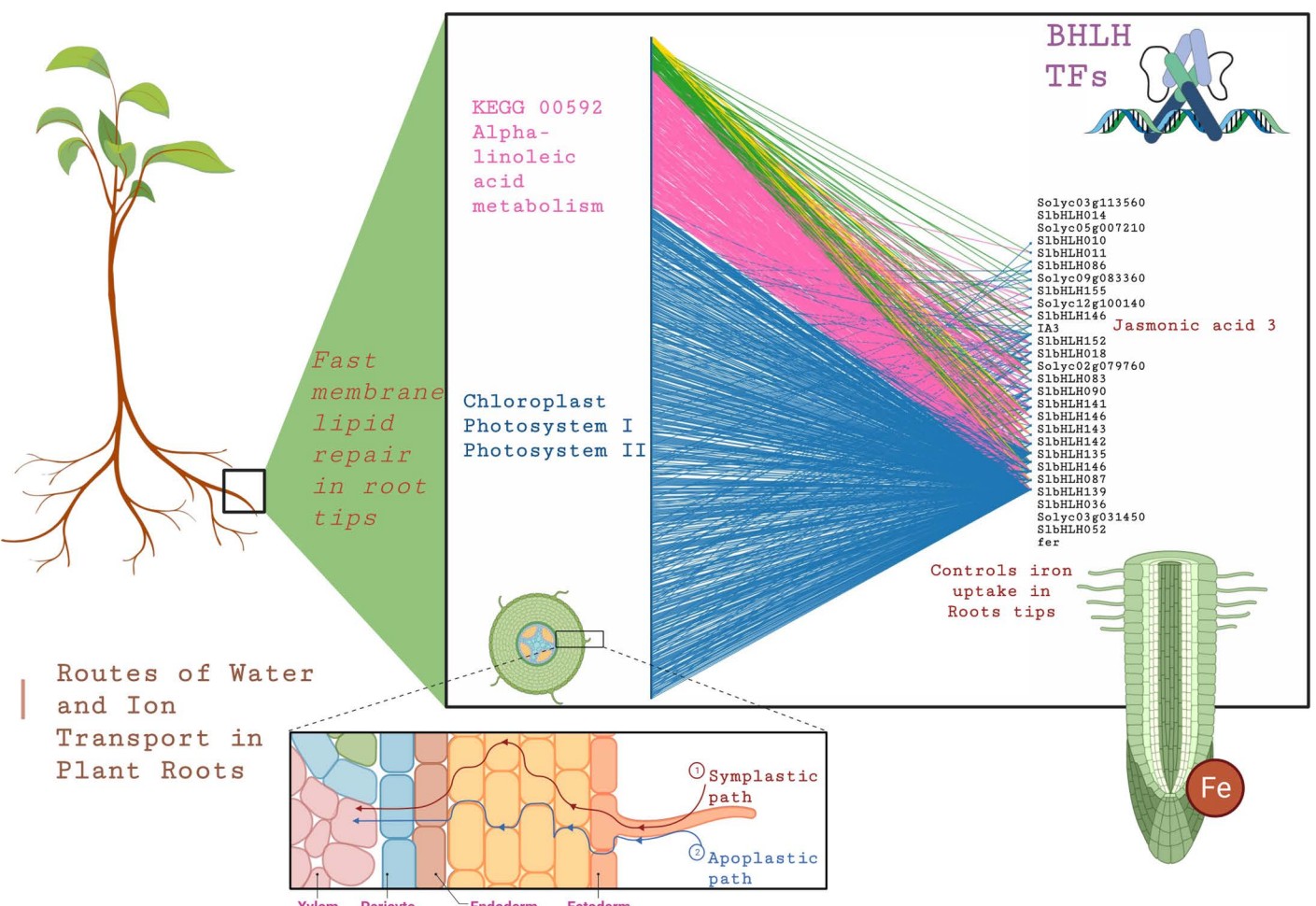

**Fig 2. Deciphering Root-Specific Regulatory Patterns.** Bipartite network analysis unveils connections between bHLH (right) TFs and target genes (left). Connections-colors represent the number of bHLH TFs for each target gene as follows: blue for one, pink for two, green for three, and yellow for four. While functional enrichment analysis highlights specific pathways, including chloroplast, Photosystem I and Photosystem II, and KEGG 00592 Alpha-linoleic acid metabolism.

non-photosynthetic tissues, such as roots, under certain conditions. For example, roots might activate chloroplast-related pathways during stress responses, such as viroid infection, reflecting an evolutionary remnant of gene regulation or crosstalk between metabolic and stress pathways. For instance, the bHLH-Jasmonic Acid 3 *Solyc08g076930.1* (IA3) is a key regulator within this network, controlling the plant's response to pathogens through the accumulation of jasmonic acid, a primary defense mechanism activated by plants upon pathogen infection, [47].

### 3.1.2. Insights from the leaf bipartite network.

The leaf bipartite networks identify interactions between 30 bHLHs and 1970 target genes. These genes are categorized based on the number of TFs regulating them: 715 targets regulated by a unique bHLH, 874 targets regulated by two bHLHs, 336 regulated by three, 40 regulated by four, and 5 targets regulated by more than four bHLHs. Genes regulated by more than four regulators are only observed in the leaf tissue-specific network. Much like its presence in the root, *Solyc08g076930.1* IA3 is also a regulator in the leaf network, underscoring its significance as a key player in plant responses to pathogen infections. Functional enrichment analysis shows that the *KEGG:00073 Cutin, suberine, and wax biosynthesis* is regulated by unique TFs, while the aspartic-type

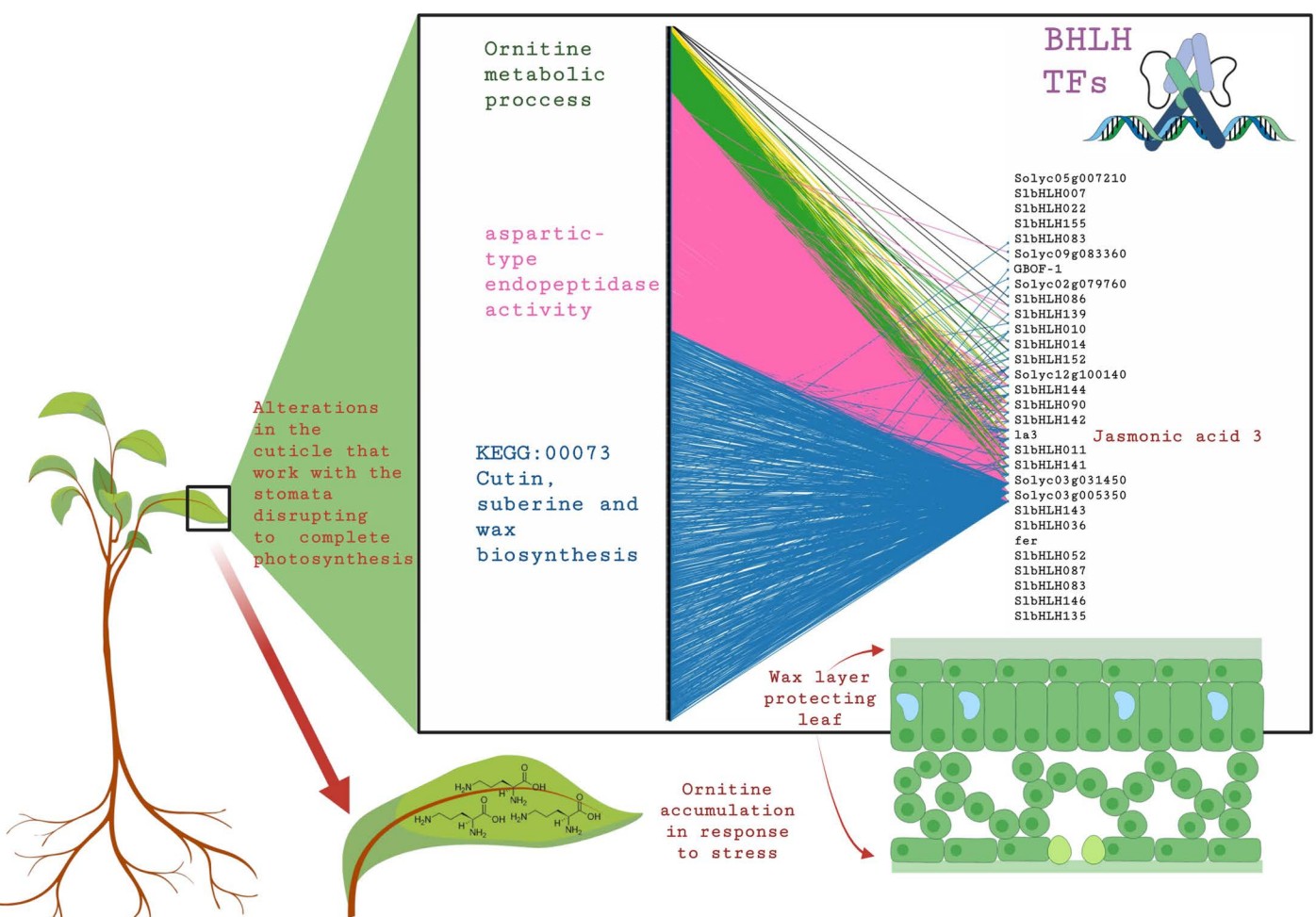

**Fig 3. Deciphering Leaf-Specific Regulatory Patterns: bipartite network analysis unveils connections between bHLH (right) TFs and target genes (left).** Connection color represents the number of bHLH regulation of each target gene as follows: blue for one, pink for two, green for three, and yellow for four. While functional enrichment analysis highlights specific pathways, including cutin, suberine, and wax biosynthesis, endopeptidase activity, and ornithine metabolic processes, the leaf-specific network showcases the exclusive presence of genes regulated by more than four TFs.

endopeptidase activity has two regulators, and ornithine metabolic processes have genes with three regulators, as illustrated in **Fig 3**.

The presence of genes regulated by a unique bHLH indicates highly specialized control, essential for processes such as cutin, suberine, and wax biosynthesis (KEGG:00073), which are critical for forming protective barriers against pathogens. Genes regulated by two bHLHs, such as those involved in aspartic-type endopeptidase activity, suggest a need for a more robust regulatory mechanism, likely to ensure precision and redundancy in defense-related proteolytic processes. Moreover, genes with three regulators, like those in ornithine metabolic processes, exhibit a higher level of regulatory complexity, potentially reflecting the importance of these pathways in the plant's metabolic adjustments during stress responses. The observation that genes regulated by more than four TFs are exclusive to the leaf tissue-specific network underscores the intricate and tissue-specific regulatory dynamics required for effective pathogen defense.

### 3.2. Dissecting gene co-expression modules and their distinct regulatory patterns in root tissue

Shifting from our broad analysis of bipartite networks, we now delve into a detailed examination of each co-expression module behavior across different plant tissues. In our analysis of root tissue, we identified ten distinct gene co-expression modules, labeled M1 to M10, with sizes ranging from 23 to 216 genes. We selected the modules that showcased distinctive regulatory dynamics and potential functional implications across the different experimental conditions, as described in Section 2.5, **Fig 4**.

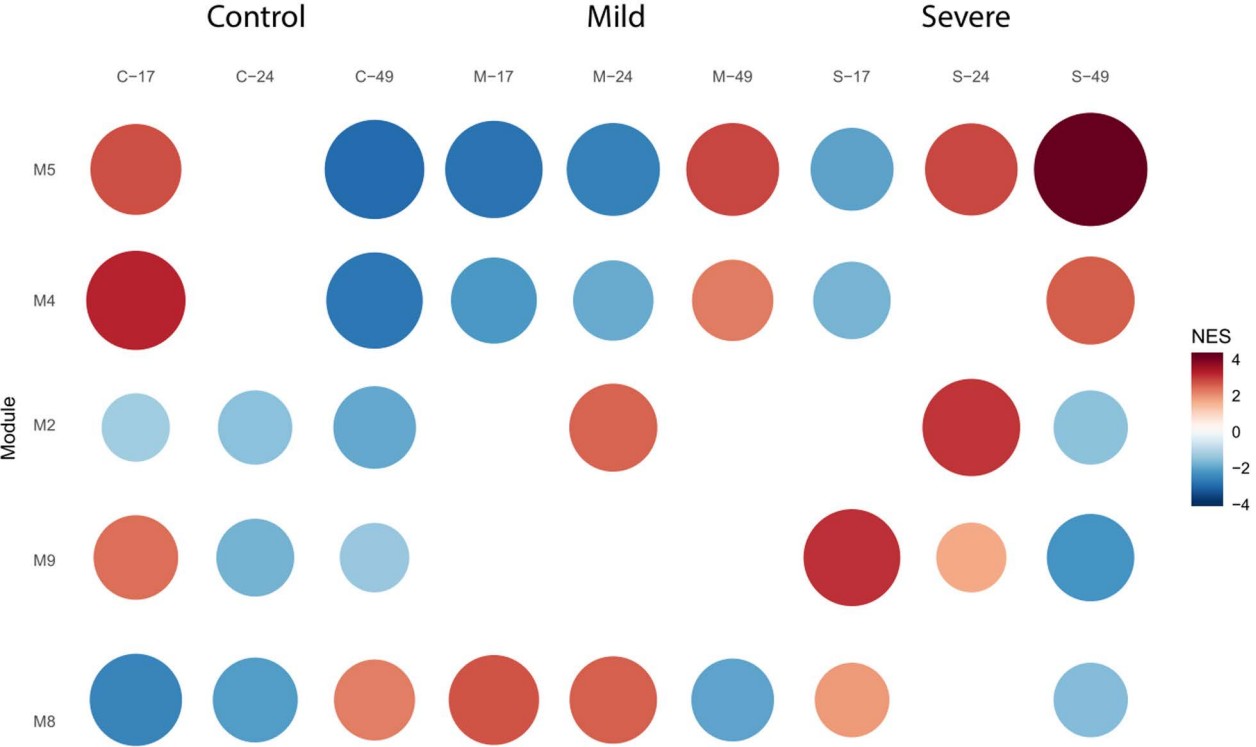

**Fig 4. Dynamics of Co-Expression Modules in Root Tissue.** GSEA was employed to unveil the module activity across distinct conditions, including control, PSTVd M, and PSTVd S23 strains, throughout the time course analysis of root tissue. Notably, the M5, M4, M2, M9, and M8 modules emerged as highly significant in this context, showcasing distinctive regulatory dynamics and potential functional implications across the different experimental conditions.

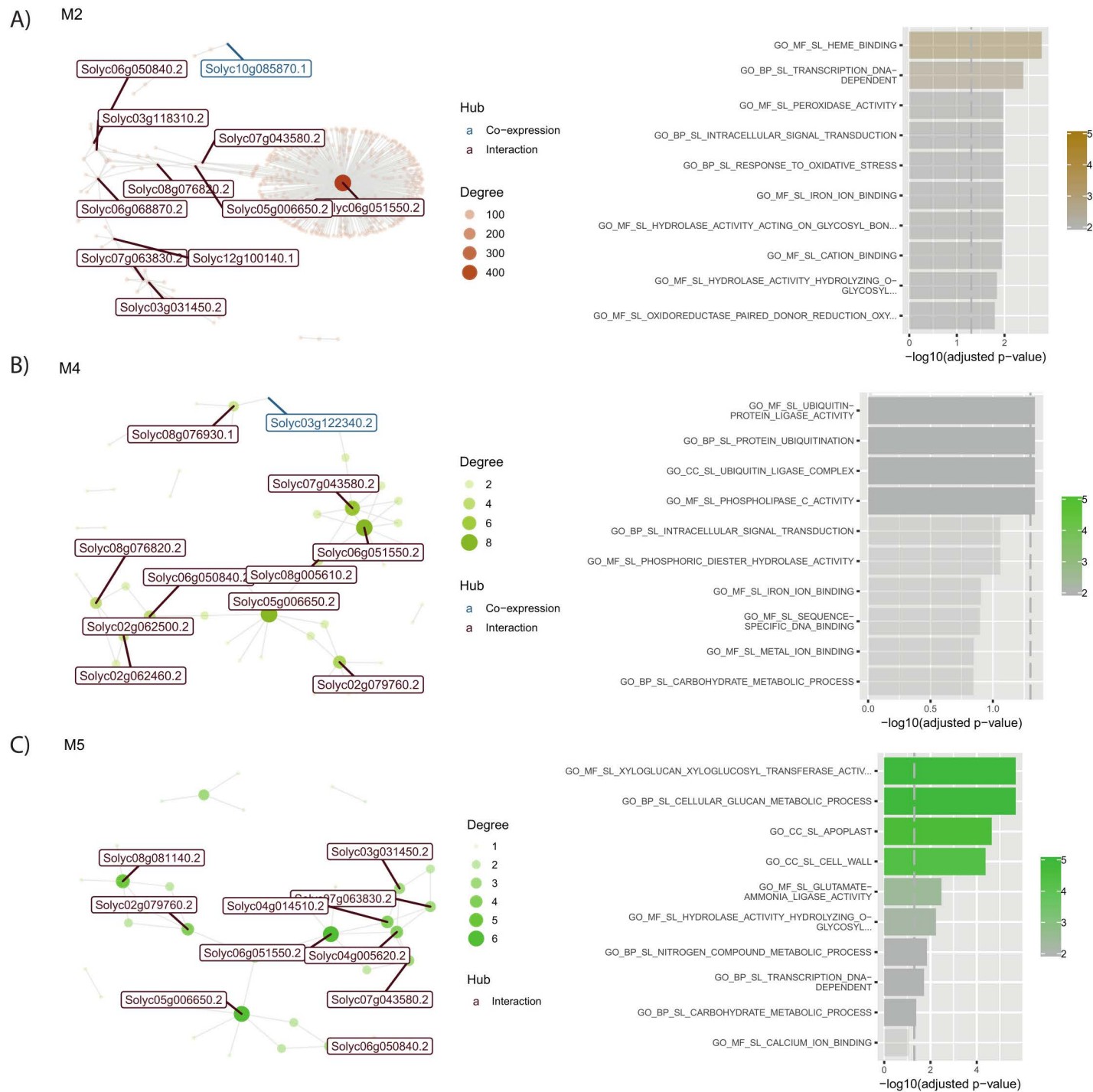

**Fig 5. Co-expression interaction network and functional enrichment of modules firstly repressed and then overexpressed in root.** Each node on the network represents a gene, size is proportional to the number of interactions (node degree). Hub genes are listed with their names, in red if they only interact, and in blue if they are co-expressed with their targets. Next to each network, biological functions of their genes are shown. Length of the bar represents significance, and color gradient number of genes implicated in each GO Term. Interaction network and biological functions for **A)** M2; **B)** M4; and **C)** M5.

**3.2.1. Modules that exhibit strong symptom induction in response to the severe strain.** Three modules, M2, M4, and M5, showed significant regulatory shifts during the 24-day symptom development phase of the severe PSTVd strain, as illustrated in **Fig 5A**. These modules displayed a coordinated surge in gene expression **Fig 5B** as root symptoms—such as stunted growth, root necrosis, and reduced root mass—worsened (24 dpi). These synchronized expression patterns could suggest that these genes may play a role in the plant's stress response and the progression of these symptoms.

Root tissue analysis identifies bHLH-guided regulatory processes such as heme-binding (M2), ubiquitin pathways (M4), and cell wall dynamics (M5) as central during symptom development under the severe strain. These biological events are synchronized with enhanced gene expression, suggesting the plant's complex adaptive response, which encompasses not only defense but also critical functions like stress tolerance, growth regulation, and metabolic balancing. Hub genes, including *Solyc06g051550* (heme-binding) and *Solyc05g006650.2* (SlbHLH036), act as central regulatory nodes. Their dense interconnectivity suggests they integrate signals from various pathways, acting as amplifiers of the plant's coordinated response to diverse stressors, not limited to pathogen defense but also playing roles in maintaining cellular homeostasis and structural integrity during stress conditions. This regulatory role highlights their importance in balancing multiple physiological processes essential for plant resilience.

**3.2.1.1 M2: Coordinated responses associated to Fe inefficient bHLH transcriptional regulator** Module 2, comprising 123 genes, underscores the integral role of the bHLH transcriptional regulator, *Solyc06g051550* (gene symbol = SlbHLH036), in modulating iron uptake efficiency, this TF present interaction with more than 400 genes in this module, which points out its importance. The GSEA as illustrated in **Fig 5A**, denotes a marked upregulation during symptom proliferation in both PSTVd infection intensities, implying Module 2's cardinal role throughout the infection trajectory. Notably, this regulation is predominantly discernible within the epidermal cells and marginally within the outer cortical cell strata of root apexes. The gene regulator exhibits an affinity for heme binding, hinting at its potential involvement in iron metabolism regulation. Simultaneously, *Solyc10g085870.1* (depicted in blue in **Fig 5A** is co-expressed in the network. This gene encodes a UDP-glycosyltransferase 73C3 that plays a crucial role in glycosylation processes, pivotal for modulating the metabolism of myriad cellular compounds, [48].

**3.2.1.2 M4: Lipoxygenase D as a key player in defense processes** Module 4, containing 79 genes, is directed by its central hub, *Solyc05g006650* (SlbHLH036). Among its co-expressed genes is the critical *Solyc03g122340.2*, Lipoxygenase D or LOX-D, highlighted in **Fig 5B**. This enzyme is vital in oxylipin production, influencing a variety of plant functions, from growth to defense. Specifically, LOX-D, part of the lipoxygenase pathway, oversees the creation of various oxylipins. The compounds from LOX-D are especially crucial in plant-pathogen dynamics and interplant signaling, [49,50]. This module accentuates LOX-D's central role in producing oxylipins essential for plant defense and growth.

**3.2.1.3 M5: Cell wall dynamics and divergent strain responses** Module 5, which comprises 75 genes, is predominantly enriched in cell wall-associated processes. Two bHLH TFs *Solyc06g051550* and *Solyc05g006650* (SlbHLH036), serve as key central connectors within this module. The expression dynamics within this module exhibit strain-specific variations, especially between the two PSTVd strains. The M-strain showcases gene repression during heightened symptom manifestation, in contrast to the S-strain which depicts gene induction, underscoring differential strain responses. However, both strains evince a unison in gene induction during the recovery phase, alluding to shared recovery mechanisms post-infection.

**3.2.2. Modules that exhibit early symptom induction and recovery phase repression in response to the severe strain.** These identified gene modules showcase a distinct pattern of gene expression in response to the severe strain of infection. During the early stages of infection, a synchronized induction of gene expression is observed, indicating the plant's immediate response to the stressors induced by the severe strain of PSTVd, **Fig 6**. However, as the infection progresses and the plant enters the recovery phase, a notable shift occurs.

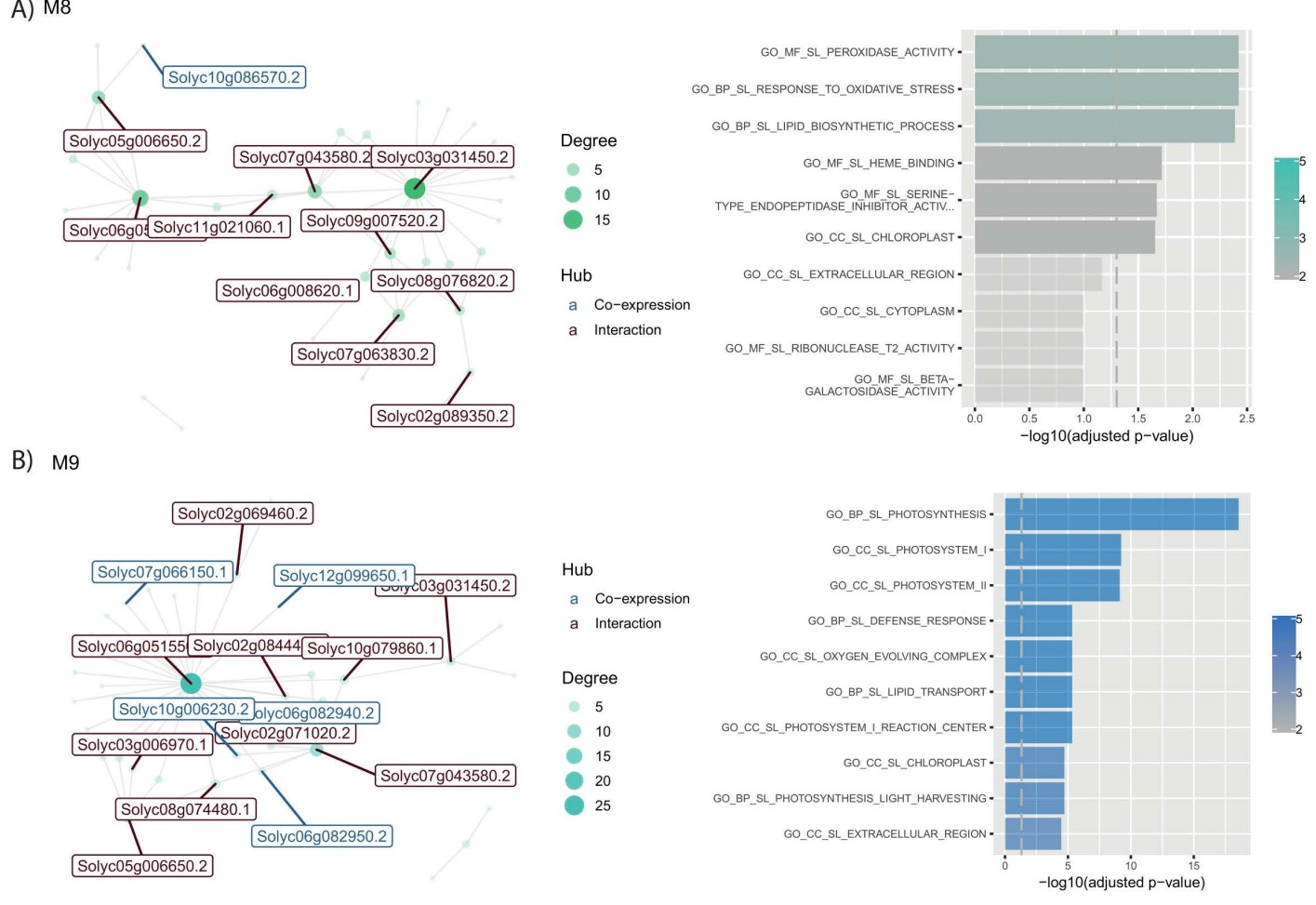

**Fig 6. Co-expression network and functional enrichment of modules at early symptoms and repressed at recovery in root-tissue.** Each node on the network represents a gene, size is proportional to the number of interactions (node degree). Hub genes are listed with their names, in red if they only interact, and in blue if they are co-expressed with their targets. Next to each network, biological functions of their genes are shown. Length of the bar represents significance, and colour gradient number of genes implicated in each Gene Ontology Term. **(A)** Interaction network and biological functions for M8; **(B)** Interaction network and biological functions for M9.

**3.2.2.1 M8: Unravelling oxidative stress response and lipid biosynthesis dynamics**  Module 8, consisting of 49 genes, focuses on oxidative stress response and lipid formation. The hub gene, *Solyc03g031450*, interacts with 15 distinct genes. Meanwhile, Palmitoyltransferase PFA4 (*Solyc10g086570*), linked with endoplasmic reticulum and Golgi apparatus formation, co-expresses with hub genes in this module,[51]. Throughout the infection by both PSTVd strains, a unified surge in gene expression occurs during pronounced symptom development, emphasizing the plant's defensive measures against PSTVd stress. However, this amplification is counteracted by gene suppression during the recovery, hinting at a complex regulatory play in oxidative stress response and lipid biosynthesis in the context of PSTVd pathogenesis.

**3.2.2.2 M9: Unmasking Photosynthesis and Defense Response Dynamics**  Module 9 comprises 43 genes linked to photosynthesis and defense. These genes exhibit varying activity between healthy and PSTVd-S infected plants. Specifically, there is an uptick in gene activity during early infection stages, which reverts during recovery. Particularly, the genes within this module display induction during the first and second stages of infection progression, aligning with

symptom development. However, a notable departure from this pattern occurs during the recovery stage, characterized by gene repression. This significant shift in gene expression dynamics emphasizes the intricate interplay between photosynthesis and defense responses during PSTVd infections. In this module, five primary hub genes are identified in the root, all linked to photosynthesis. These include photosystem I reaction center subunit V (*Solyc07g066150*); photosystem II 5 kDa protein, chloroplastic (*Solyc12g099650*); chlorophyll a-b binding protein 7, chloroplastic (*Solyc10g006230*) and two genes for Photosystem I reaction center subunit XI (*Solyc06g082940* and *Solyc06g082950.2*). Their co-expression aligns with the module's focus on photosynthesis and the photosystem. Additionally, *Solyc06g051550 Fe inefficient bHLH transcriptional regulator* is also highlighted as a key regulator in this module, like its function in other modules.

### 3.3. Exploring leaf tissue gene co-expression modules

Shifting our focus to leaf tissue, our exploration highlights gene co-expression patterns that elucidate the plant's molecular reactions in the presence of PSTVd. Our research identifies seven distinct gene co-expression modules (M1 to M7) within the leaf tissue, **Fig 7**. These modules, ranging in size from 24 to 331 genes, embody the complex interplay of genes and regulators reacting to the pathogen. Specifically, modules M5, M1, M3, M6, and M2 emerge as particularly noteworthy due to their pronounced regulatory shifts across conditions.

#### 3.3.1. Shared regulatory behavior: a consistent pattern.
Notably, four of these modules —M1, M3, M5, and M6— exhibit a remarkably similar pattern of behavior. In control plants, a consistent trend of induction is observed across almost all stages of the study. This underscores the essential role of these modules in normal cellular processes and

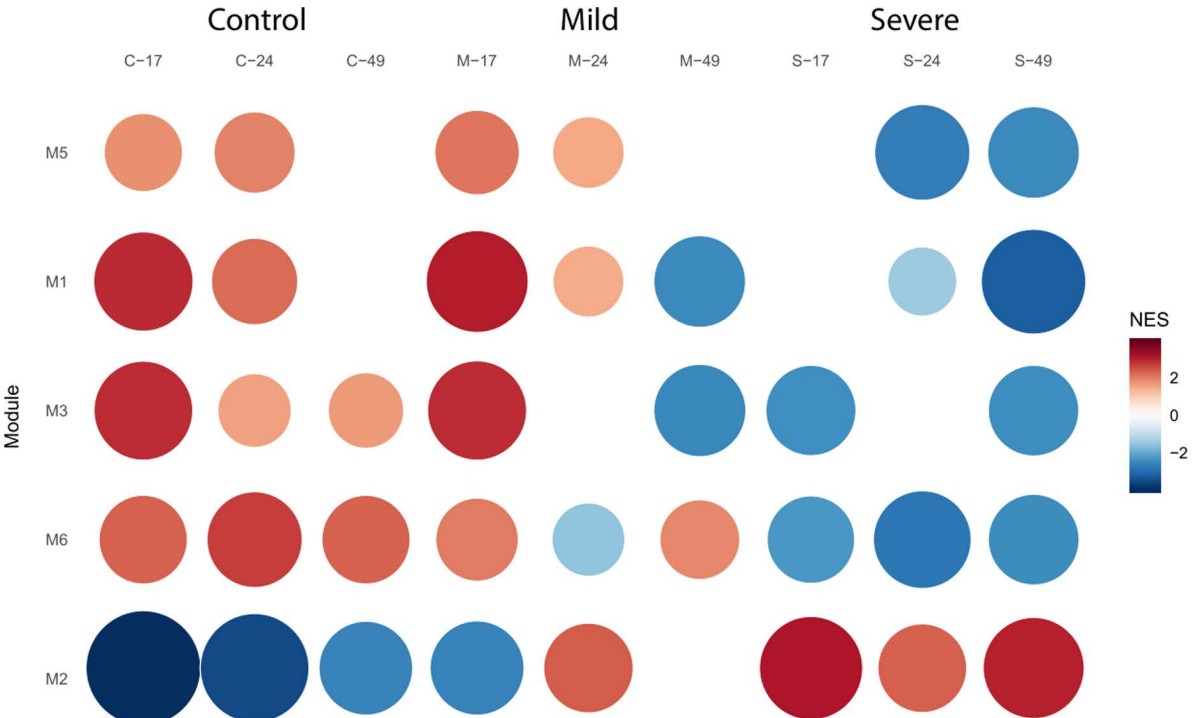

**Fig 7. Dynamics of Co-Expression Modules in Leaf Tissue.** GSEA for the significant co-expression modules (M1, M2, M3, M5, and M6) within leaf tissue. It visually represents the module activity under control, PSTVd M, and PSTVd S23 strains during the time course conditions. A visual representation captures the regulatory dynamics of the five key co-expression modules (M5, M1, M3, M6, and M2) within leaf tissue. The figure illustrates the contrasting behavior of these modules in control plants and those infected with PSTVd-S.

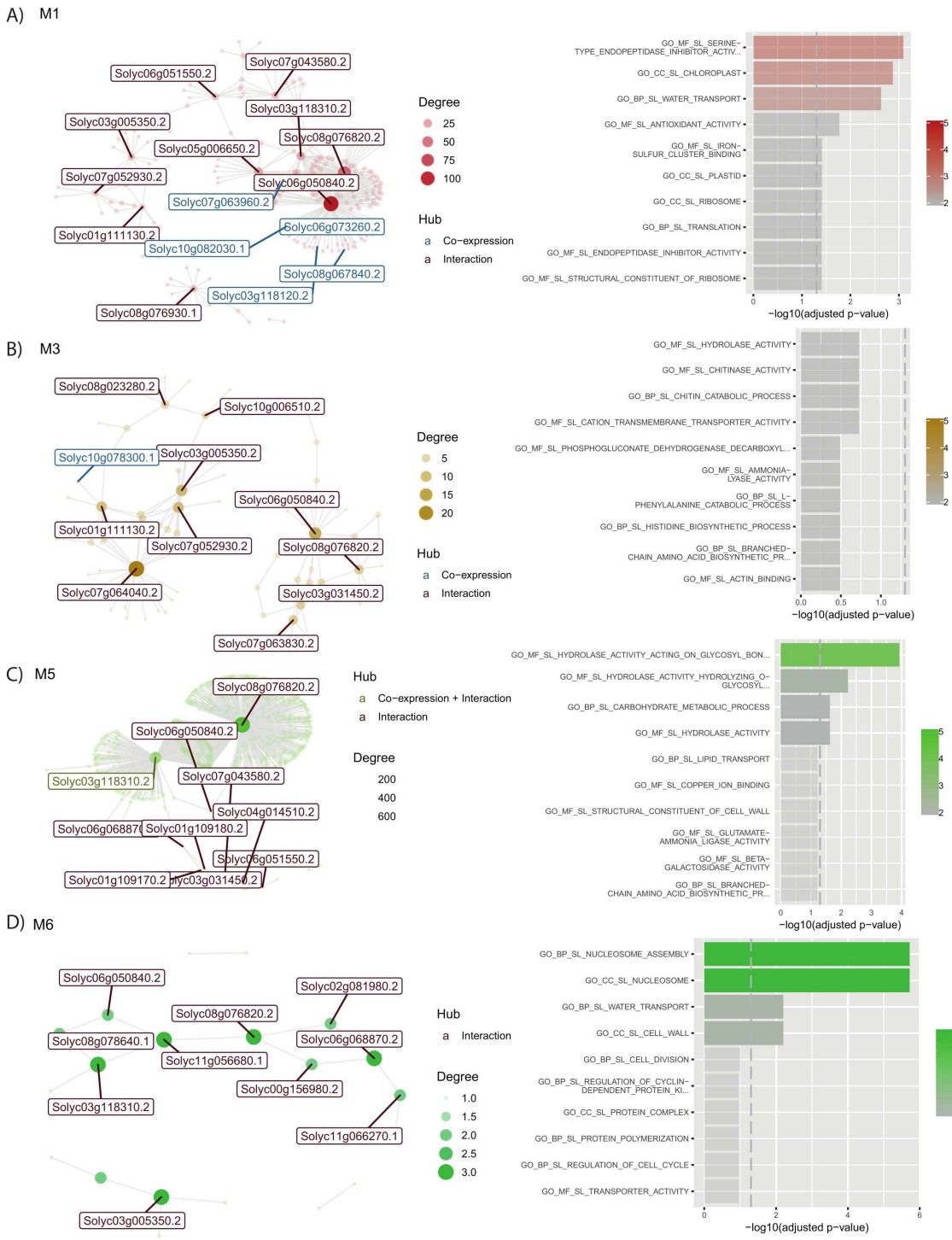

**Fig 8. Co-expression network and functional enrichment of modules repressed at recovery with PSTVd-S23 in leaf.** Each node on the network represents a gene, size is proportional to the number of interactions (node degree). Hub genes are listed with their names, in red if they only interact, and in blue if they are co-expressed with their targets. Next to each network, biological functions of their genes are shown. Length of the bar represents significance, and color gradient number of genes implicated in each GO Term. **(A)** Interaction network and biological functions for M1; **(B)** Interaction network and biological functions for M3; **(C)** Interaction network and biological functions for M5; **(D)** Interaction network and biological functions for M6.

homeostasis. However, contrast emerges when examining PSTVd-S infected plants. These modules, which normally show induction in control plants, exhibit a notable trend of repression across most studied stages, <u>Fig 8</u>.

### 3.3.1.1 M1: Orchestrating chloroplast biogenesis

Module 1, comprising 331 genes has five co-expressed genes—*Solyc07g063960.2* (50S ribosomal protein L24), *Solyc06g073260.2* (NAD-dependent epimerase/dehydratase), *Solyc10g082030.1* (Peroxiredoxin), *Solyc08g067840.2* (PsbP domain-containing protein 5, chloroplastic), and *Solyc03g118120.2* (Transferase transferring glycosyl groups). These genes exert influence over a range of interactions, spanning from 25 to 100, highlighting their pivotal roles within the module. Additionally, SlbHLH135 (*Solyc06g050840*) is a hub gene having more than a hundred interactions. The genes within Module 1 encompass a diverse array of functions, shedding light on the multifaceted roles they play within leaf tissue. In chloroplast biogenesis, these genes contribute to vital processes, ensuring the plant's capacity to harness light energy for growth and sustenance, <u>Fig 8A</u>.

### 3.3.1.2 M3: Exploring the landscape of metabolic regulation

Within Module 3, comprising 134 genes, parallels can be drawn with Module 1's behavioral dynamics. Central to this module is the co-expressed hub-gene *Solyc10g078300.1*, associated with single-stranded nucleic acid binding R3H protein, emphasizing its role in metabolic regulation. Moreover, the Photosystem I reaction center subunit V (*Solyc07g064040*) emerges as a pivotal regulator, interacting extensively with over 20 distinct genes. The influence of this gene assembly extends to various biomolecules such as chitinase, phenylalanine, and histidine, <u>Fig 8B</u>.

### 3.3.1.3 M5: Hydrolase activity regulation by solyc03g118310 bHLH083 in leaf tissue dynamics

Module 5 is regulated by the bHLH TF *Solyc03g118310* (gene symbol = bHLH083), a pivotal gene in the co-expression network associated with hydrolase activity. This module's atypical repression across developmental stages diverges from its expected induction in control conditions, intimating a sophisticated regulatory role of bHLH083 in plant development and defense. Hydrolases, central to Module 5, catalyze the degradation of biomolecules, thus modulating metabolic pathways, cell wall architecture, and signaling networks[52,53]. The observed regulatory pattern suggests a potential energy-conservation strategy, positioning the plant for rapid defensive action against environmental stressors. The orchestrated expression of Module 5 under bHLH083's aegis likely represents a strategic standby mode, priming the plant for expedited response upon pathogen detection, <u>Fig 8C</u>.

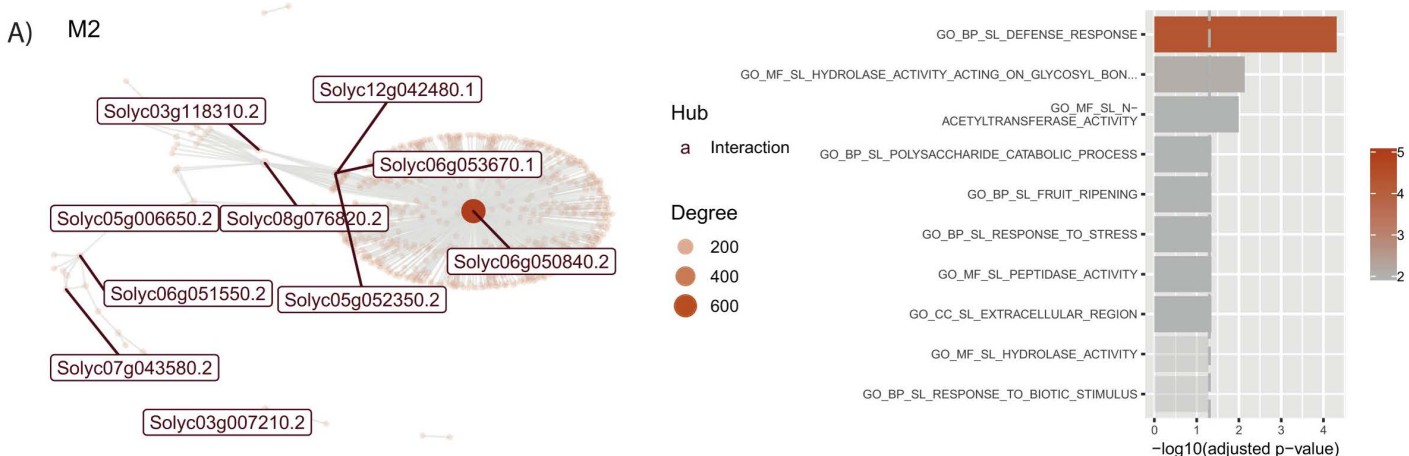

**Fig 9. Co-expression network and functional enrichment of module 2 in leaf.** Each node on the network represents a gene, size is proportional to the number of interactions (node degree). Hub genes are listed with their names, in red if they only interact, and in blue if they are co-expressed with their targets. Next to each network, biological functions of their genes are shown. Length of the bar represents significance, and color gradient number of genes implicated in each Gene Ontology Term. Interaction network and biological functions for M2.

**3.3.1.4 M6: Cell division and beyond**   Module 6, with 28 co-expressed genes, navigates the critical landscape of cell division. Its genes, entwined in processes such as nucleosome assembly, nucleosome as a cellular component, and the very essence of cell division, unveil a tightly orchestrated performance within the leaf tissue, **Fig 8D**.

**3.3.2.  M2 shows a dynamic contrast in defense.**  In contrast to other modules, Module 2, with 149 genes, exhibits a unique response to PSTVd infection, **Fig 9**. Furthermore, a hub gene with more than 600 interactions is shown for this module, SlbHLH135 (*Solyc06g050840*). While other modules are induced, genes in Module 2 are repressed in control samples, suggesting a readiness for defense. However, during PSTVd-S infection, a strong gene activation occurs across multiple stages.

### 3.4.  Mapping the regulatory landscape: a bridge between modules

Biological network motifs are recurring, significant patterns of interconnections that appear more frequently in real networks than in randomized ones, [46]. Within the complex landscape of gene interactions highlighted in our study, the bifan motif stands out as a prominent regulatory beacon. This motif, a distinctive architecture of regulatory interactions, plays a central role in deciphering the sophisticated governance of gene regulation within our co-expression modules. The bifan motif is emblematic of a specific regulatory paradigm, distinguished by its two primary *"source"* nodes, represented by bHLH TFs, and two secondary *"target"* nodes, which are the genes under regulation (**Fig 10**).

A defining feature of this configuration is the independent regulation by each source node of both target nodes. The structure of the bifan motif enables a coordinated regulatory response, wherein alterations in the activity or levels of

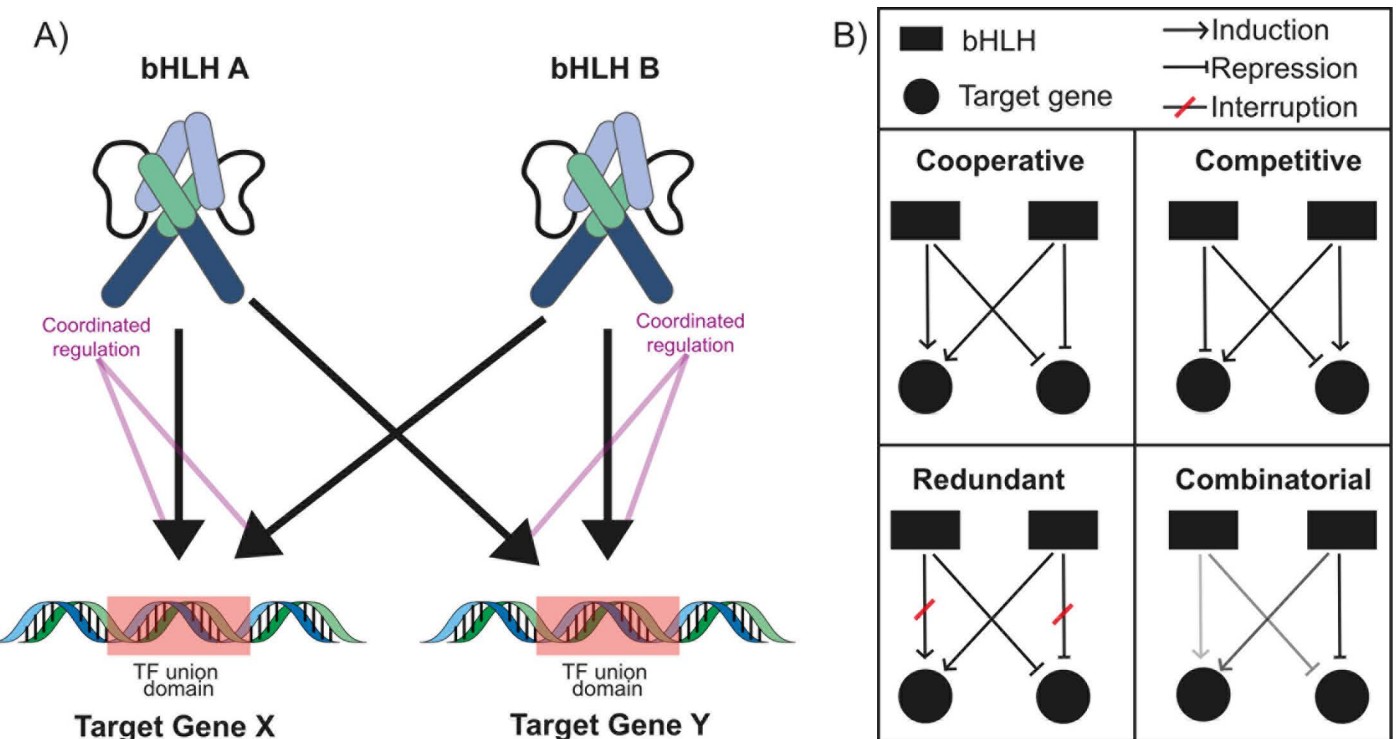

**Fig 10.  Bifan Motif and its functionality in gene regulation. (A)** Graphic illustration of the Bifan motif: two different bHLH regulating the expression of two different target genes; arrows depict regulation; **(B)** Examples of Bifan motif activity in gene regulation. Legend for the diagram is shown at the top. Four different regulation types are shown: Cooperative, Competitive, Redundant, and Combinatorial. The color gradient indicates level of regulation: the darker, the stronger.

nodes A and B correspondingly modulate the expression or activity of nodes X and Y, as visualized in **Fig 10A**. Beyond its unique dual-regulatory characteristic, the bifan motif facilitates an intricate synergy of transcriptional control within our identified gene modules. Crucially, this motif serves as a linchpin, bridging separate modules and revealing convergent and divergent regulatory patterns that intricately weave through the vast tapestry of gene interactions, [54,55]. The bifan motif underscores several pivotal attributes such as: 1) *Flexibility,* the dual regulatory nodes confer a degree of adaptability, enabling the system to recalibrate in response to nuanced environmental changes; 2) *Robustness:* built-in redundancy ensures resilience against disruption, safeguarding the integrity of regulatory processes; and 3) *Signal Integration:* adept at amalgamating signals from disparate pathways, enabling the cellular machinery to craft a unified and coherent response, **Fig 10B**. The prevalence of the bifan motif within our co-expression networks deciphers the regulatory intricacies governing the plant's response to PSTVd infection. This motif's role as a nexus of crosstalk underscores its significance in shaping the complex interplay of gene expressions, shedding light on the molecular dialogues that govern the defense and adaptation strategies of the host plant.

## 4. Discussion

Our study deepens our understanding of the complex regulatory framework of tomato response to PSTVd infection by analyzing gene co-expression modules, with a specific focus on the role of bHLH TFs in root and leaf tissues. Unlike traditional transcriptome analyses that primarily highlight differential gene expression, our approach sheds light on the regulatory interactions and network properties that are key to understanding plant defense mechanisms.

Notably, co-expression networks exhibit a complex regulatory structure where the bifan motif is the most prominent. This motif acts as a primary regulatory coordinator, allowing multiple transcription factors, such as bHLHs, to control the expression of several target genes simultaneously. In the bifan motif, two regulators co-regulate two target genes, creating a tightly connected module where changes in the activity of one regulator can propagate through the network, leading to coordinated gene expression. The dominance of the bifan motif highlights the highly organized molecular interactions within these networks, suggesting that during pathogen challenges, there is precise integration of signals that modulate the plant's defense response [55]. The GO analysis further supports this, identifying key functional categories such as stress response, energy metabolism, and protein modification, which are significantly enriched in the identified gene modules. This enrichment underscores the broad functional role of these modules in orchestrating the plant's systemic response to PSTVd-induced stress, integrating regulatory signals to optimize both defense and survival strategies.

Specifically, in root tissues, functional enrichment analysis identified key bHLH-driven regulatory processes such as heme-binding, ubiquitin defense, and cell-wall dynamics, which play significant roles in symptom development under severe strain conditions. Hub genes like *Solyc06g051550* (heme-binding) and *Solyc05g006650.2* (SlbHLH036) are central to stress adaptation, participating in critical functions like glycosylation and ubiquitin defense, which help maintain cellular homeostasis and activate defense mechanisms, including jasmonic acid biosynthesis [52,53,56–58]. This analysis extends previous research by highlighting the specific regulatory influence of bHLH TFs, revealed through co-expression network analysis [18]. The identified genes in these modules likely contribute to root symptoms, such as stunted growth and root necrosis, through pathways involved in stress adaptation and metabolic disruption. Genes linked to membrane stability, energy metabolism, and nutrient transport may drive these stress responses, while metabolic disruptions impair root development and exacerbate viroid symptoms, indicating these genes' active role in defense mechanisms.

Interestingly, photosynthesis-related genes (e.g., *Solyc07g066150, Solyc12g099650*) were also found in root tissues, coordinated by bHLH TFs like *Solyc03g031450* and *Solyc10g086570*. While typically associated with photosynthesis in leaves, these genes may have non-photosynthetic roles in roots, potentially involved in redox regulation and secondary metabolism under viroid stress. Indeed, chloroplast genes, generally active in leaves, have been observed in roots during stress conditions, such as shoot removal. Genes like GOX1 and GLDP1 are upregulated in roots in response to cytokinin signaling, suggesting they contribute to redox regulation and metabolic adjustments in non-photosynthetic tissues[59,60].

This behavior could represent an evolutionary remnant of gene regulation, where chloroplast-related genes, once solely for photosynthesis, retain secondary functions and now aid plant stress responses across various tissues[61,62].

In leaf tissues, key bHLH genes such as *Solyc06g050840* (SlbHLH135) play a critical role in regulating processes like chloroplast formation and response to abiotic stress, including photosynthesis under fluctuating environmental conditions. These TFs are involved in maintaining the balance of chloroplast biogenesis and repair, especially during stress responses like pathogen attack or environmental changes. Similarly, *Solyc10g078300.1* is associated with metabolic stability, ensuring proper energy management during infection or stress conditions [63,64]. The hub gene SlbHLH083 regulates hydrolase activity and carbohydrate metabolism, which are essential for energy homeostasis, particularly during viroid infections, where metabolic adjustments are critical for both defense and survival. The involvement of ribosomal and chloroplast-related genes highlights the importance of maintaining translation and chloroplast functionality under stress, further contributing to a comprehensive bHLH-regulated network that coordinates multiple physiological processes.

Comparative analysis reveals distinct bHLH-driven regulatory patterns between root and leaf tissues Interestingly, root modules focus on photosynthesis-related processes, while leaf modules also emphasize structural integrity and defense. The identification of most of the bHLH genes across both networks suggests a universal defense strategy, with bHLH TFs functioning as versatile regulators of plant responses to viroid infection. For the role of bHLH-Jasmonic IA3 (*Solyc08g076930.1*) in modulating the jasmonic acid (JA) pathway, recent studies have highlighted the critical role that bHLH-TFs play in regulating this pathway, which is central to both biotic and abiotic stress responses in plants. JA acts as a key signaling molecule involved in the regulation of plant defense mechanisms against herbivores and pathogens, and also influences growth processes like root elongation and senescence. bHLH TFs, particularly MYC2, have been shown to directly interact with JA-responsive genes to enhance the plant's defense signaling efficiency, [65]. This enhances the plant's ability to respond to multiple stressors simultaneously by integrating JA signaling with other pathways, such as those related to auxin and abscisic acid [65–69]. Additionally, it has been specifically implicated in the modulation of the JA signaling pathway, further emphasizing the broad role that bHLH TFs play in orchestrating plant responses to environmental challenges. By coordinating the expression of genes involved in carbohydrate metabolism, energy balance, and cell wall modification, IA3 plays a vital role in fine-tuning defense mechanisms during viroid infections.

The distinct activity of bHLH TFs in roots and leaves during PSTVd infection underscores their versatility in managing tissue-specific stress responses. Bipartite network analysis in this study reveals complementary regulatory dynamics in these organs, highlighting their specialized roles in stress adaptation.

For instance, in roots, GO analysis reveals the activation of chloroplast-associated pathways, including Photosystems I and II, despite the non-photosynthetic nature of this tissue. As previously discussed, this activation likely represents an evolutionary remnant of gene regulation or a form of metabolic crosstalk, allowing roots to leverage chloroplast-related pathways to maintain redox homeostasis and redirect energy metabolism toward survival mechanisms. Key regulators such as SlbHLH052 Fer involved in heme binding and iron homeostasis, illustrate the critical importance of structural resilience and metabolic adaptation in root stress responses to viroid infection.

In contrast, leaves exhibit a more complex regulatory network, involving 30 bHLH TFs and 1970 target genes. A unique feature of the leaf network is the presence of genes regulated by more than four bHLH TFs, highlighting the intricate coordination required to balance energy homeostasis and defense mechanisms. Leaf-specific bHLH TFs, such as GBOF-1 (*Solyc06g072520.1*) and SlbHLH022 (*Solyc03g097820.1*) prioritize metabolic regulation, and secondary metabolite biosynthesis to facilitate rapid responses to biotic stress.

The modular co-expression analysis reveals distinct tissue-specific dynamics in the plant's response to viroid infection, with root and leaf hub genes exhibiting entirely unique profiles. This specialization underscores the plant's systemic strategy to deploy tailored molecular mechanisms to address the specific environmental and metabolic challenges faced by each organ.

In roots, bHLH TFs integrate with JA and ethylene signaling pathways to regulate nutrient uptake and maintain structural stability, ensuring effective stress adaptation. For example, *Solyc06g034370.1* hub gene (Pectinesterase) modifies the cell wall, providing enhanced mechanical support and increasing root resilience during viroid infection. Additionally, ethylene-responsive genes, such as *Solyc09g075420.2* (Ethylene-responsive transcription factor 2b), play a pivotal role in integrating ethylene signaling to further support redox balance and structural integrity.

In contrast, leaf-specific hub genes, such as *Solyc06g073260.2* (NAD-dependent epimerase) and *Solyc10g078300.1* (Single-stranded nucleic acid binding R3H protein), prioritize immune signaling and secondary metabolite biosynthesis. These processes effectively reallocate metabolic resources to strengthen pathogen resistance, aligning with salicylic acid (SA)-dependent systemic acquired resistance (SAR) and enhancing both localized and systemic defense mechanisms. The interplay of bHLH TFs with key hormonal pathways, including JA, ethylene, auxin and SA, refines tissue-specific stress responses by coordinating distinct molecular and physiological processes.

In roots, modules associated with severe PSTVd infection at 24 dpi highlight key stress-adaptive pathways, including ubiquitin-mediated defense, and cell wall dynamics. These modules display temporal dynamics, with early induction and subsequent repression during recovery, particularly in genes related to photosynthesis, such as photosystem I reaction center subunit V (*Solyc07g066150*) and chlorophyll a-b binding protein 7 (*Solyc10g006230*). In leaves, most modules show repression, except for Module M2, which is linked to defense responses. Modules M1 and M3 emphasize repression of chloroplast biogenesis and metabolic regulation, respectively, while Module 5 (M5), regulated by bHLH083 (*Solyc03g118310*), underscores the role of carbohydrate metabolism and energy homeostasis. Repression of chloroplast-related genes, including those involved in chloroplast biogenesis, reflects a strategic reallocation of resources from photosynthetic processes to defense, further emphasizing the prioritization of pathogen resistance in this tissue.

The differential regulation of chloroplast-related genes in roots and leaves highlights their distinct functional priorities during stress. In roots, the activation of chloroplast-associated pathways facilitates metabolic adjustments essential for stress tolerance, enabling redox homeostasis and energy redirection to survival mechanisms. In contrast, leaves repress these pathways to efficiently reallocate energy and resources toward defense, prioritizing the production of secondary metabolites and activation of immune responses. This divergence exemplifies a coordinated systemic response to viroid infection, with roots focusing on maintaining metabolic stability, while leaves concentrate on pathogen resistance and energy conservation. Moreover, these processes, including chloroplast biogenesis and repair, not only sustain photosynthetic efficiency under stress but also drive the biosynthesis of secondary metabolites such as flavonoids and alkaloids, which are crucial for pathogen defense and overall plant resilience.

Beyond the tomato-PSTVd pathosystem, bHLH TFs are integral to various plant-pathogen interactions. For example, SlBIGPETAL1 (*Solyc05g006650.2*) regulates petal morphology in tomato and has conserved roles in other species like Arabidopsis, demonstrating the evolutionary versatility of bHLH factors. In crops such as hops and sweet cherry, bHLH TFs influence key processes like metabolite production and fruit health, further underscoring their regulatory potential across different stress conditions and species [17–71].

Overall, our study highlights the pivotal roles of bHLH TFs in managing complex molecular interactions during severe PSTVd infections, regulating essential processes such as glycosylation, jasmonic acid biosynthesis, energy production, and translation. These insights advance our understanding of the tomato-PSTVd regulatory landscape and suggest broader implications for enhancing plant resilience against pathogens. Future research should focus on experimentally validating these hub genes identified in the networks to fully harness the potential of bHLH TFs in improving plant defense strategies.

## Acknowledgments

We would like to thank Prof. Matthew Hudson and Prof. Gustavo Caetano-Anolles for fruitful discussions and recommendations.

## Author contributions

**Conceptualization:** Katia Aviña-Padilla, Octavio Zambada-Moreno, Rosemarie W. Hammond, Maribel Hernandez-Rosales.

**Data curation:** Katia Aviña-Padilla, Octavio Zambada-Moreno, Marco A. Jimenez-Limas.

**Formal analysis:** Katia Aviña-Padilla, Octavio Zambada-Moreno, Marco A. Jimenez-Limas, Maribel Hernandez-Rosales.

**Funding acquisition:** Rosemarie W. Hammond.

**Investigation:** Octavio Zambada-Moreno, Marco A. Jimenez-Limas, Rosemarie W. Hammond, Maribel Hernandez-Rosales.

**Methodology:** Katia Aviña-Padilla, Octavio Zambada-Moreno, Marco A. Jimenez-Limas, Rosemarie W. Hammond, Maribel Hernandez-Rosales.

**Project administration:** Rosemarie W. Hammond.

**Resources:** Rosemarie W. Hammond, Maribel Hernandez-Rosales.

**Software:** Katia Aviña-Padilla.

**Supervision:** Rosemarie W. Hammond, Maribel Hernandez-Rosales.

**Validation:** Katia Aviña-Padilla, Octavio Zambada-Moreno, Maribel Hernandez-Rosales.

**Visualization:** Katia Aviña-Padilla, Octavio Zambada-Moreno, Marco A. Jimenez-Limas, Maribel Hernandez-Rosales.

**Writing – original draft:** Katia Aviña-Padilla, Octavio Zambada-Moreno, Rosemarie W. Hammond, Maribel Hernandez-Rosales.

**Writing – review & editing:** Katia Aviña-Padilla, Octavio Zambada-Moreno, Marco A. Jimenez-Limas, Rosemarie W. Hammond, Maribel Hernandez-Rosales.

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
