## [Decision Letter · Decision Letter 0]

29 Aug 2024

PONE-D-24-31516Dissecting the role of bHLH transcription factors in the potato spindle tuber viroid-tomato pathosystem using network approaches.PLOS ONE

Dear Dr. Hernandez-Rosales,

Thank you for submitting your manuscript to PLOS ONE. After careful consideration, we feel that it has merit but does not fully meet PLOS ONE’s publication criteria as it currently stands. Therefore, we invite you to submit a revised version of the manuscript that addresses the points raised during the review process.

We look forward to receiving your revised manuscript.

Kind regards,

Abozar Ghorbani, Ph.D

Academic Editor

PLOS ONE

**Journal Requirements:**

This research was funded by internal USDA-ARS project number 8042-22000-318-00D. K.A.-P. (CVU:227919), O.Z.-M. (CVU:1147042) and M.A.J.-L. (CVU:1035685) received financial support from the CONAHCyT. K.A.-P. had a fellowship from the Fulbright García-Robles foundation.

4. Please note that your Data Availability Statement is currently missing the repository name. If your manuscript is accepted for publication, you will be asked to provide these details on a very short timeline. We therefore suggest that you provide this information now, though we will not hold up the peer review process if you are unable.

Reviewers' comments:

Reviewer's Responses to Questions

**Comments to the Author**

1. Is the manuscript technically sound, and do the data support the conclusions?

Reviewer #1: Yes

Reviewer #2: Partly

2. Has the statistical analysis been performed appropriately and rigorously? 

Reviewer #1: Yes

Reviewer #2: N/A

3. Have the authors made all data underlying the findings in their manuscript fully available?

Reviewer #1: Yes

Reviewer #2: Yes

4. Is the manuscript presented in an intelligible fashion and written in standard English?

Reviewer #1: Yes

Reviewer #2: Yes

5. Review Comments to the Author

**Reviewer #1: ** The presented work by Katia Aviña-Padilla et al. extends previous findings on crucial role of bHLH transcription factors (TFs) in regulating gene expression during PSTVd infection in tomato. Utilizing gene co-expression network analysis, the authors have examined root and leaf tissues of the tomato during mild and severe infection. This approach has enabled them to identify the changes in gene-gene interactions that are influenced during the progression of symptom development.

Manuscript is well-structured and represents excellent work. However, it requires a series of minor modifications as outlined below:

“Ten distinct gene co-expression modules, designated as M1 to M10, have varying sizes ranging from 216 to 23.” Please clarify if the number of genes is decreasing across the modules.

“Module 2, comprising 123 genes, underscores the integral role of the bHLH transcriptional regulator, Solyc06g051550, in modulating iron uptake efficiency. This TF interacts with more than 400 genes in this module, highlighting its importance.” The phrase “400 hundred” is incorrect.

“Two BHLH genes, Solyc06g051550 and SlbHLH036, act as central connectors.” Please ensure consistent use of the term “bHLH” for genes. Also, unify the naming of SlbHLH036 as Solyc05g006650.2 for consistency. The same is applied to bHLH083.

On Fig. 5A add a, b and c to determine interaction network and biological functions for M2, M4 and M5 respectively.

In the text where you mention Fig.8 (A-D) replace the capital letters with lowcase as it is depicted in the figure.

In the legend of Fig. 10 and inside the text replace capital letter in lowcase as it is depicted in the figure.

**Reviewer #2:**  Major revisions:

1. In Page3, the second paragraph, “Yet, the presence of unknown genes in plant genomes poses challenges in data interpretation. ” It is difficult to understand the transition here. How does co-expression network analysis interpretate the data related with unknown genes? This study seems not integrate the data of any unknown genes but the known bHLH gene family. Additionally, from “This network forms a graph of…..” to the end, the introduction of co-expression network seems repetition and boring.

2. In Page 4, the 4th paragraph, why only highlight “the tomato immune response” against viroid infection? Does bHLH play major roles in the resistance to viroid infection? Additionally, the sentence “we explore their involvement in host molecular mechanisms.” seems incomplete.

3. In Page 4, last paragraph, the first sentence, these is no solid molecular and genetic evidence that can prove that “bHLH TFs regulate photosynthesis and membrane lipid repair in infected roots”. The co-expression network analysis only shows association not causal relationship.

4. Page 6, the first paragraph, why chose “Microarray datasets”? Most of the gene in microarray analysis should be known, while RNA-seq analysis can reveal some unknow genes.

5. Page 8, the second paragraph, the late sentence “This differential composition between the leaf and root……” This expression is too conclusive. No strong evidence supports “the tissue-specific regulatory roles played by bHLH TFs,”

6. Page 9, the first paragraph, generally, there is no chloroplast in root, how to explain the finding that “Genes with a single regulator are implicated in Chloroplast, Photosystems I and II,”?

7. Page 11, the first sentence, “3.2.1 Modules that Exhibit Strong Symptom Induction in Response to the Severe Strain” What are the symptoms in root of tomato plants infected with viroid? and what associations of the following identified genes in this part with these symptoms?

8. Page14, “3.3. Exploring Leaf Tissue Gene Co-Expression Modules”. The symptoms induced by viroid infection in tomato plants are obvious in leaf. Why did not analyze the module associated with symptoms in leaf like the analysis of root?

9. Page 18, third paragraph, the sentence “This synchronization is an evolutionary response to aggressive pathogens.” is difficult to understand. How to transmit to ‘evolutionary response’?

Additionally, “Notably, hub genes emerge as the linchpins, much like regulatory keystones, ensuring the efficiency of defense strategies.” Author always wants to link identified genes or gene module with plant defense. However, these genes have many biological functions, not only related to plant defense. Thus, why only highlighted plant defense?

10. Many contents of “Discussion” are repetition of “Results”. This part should focus one or several import point or problem, not mention each point relates to results. The most important point should be the discussion about the new findings or explanations about plant responses to viroid infection, especially compares with those findings obtained using transcriptome analysis, because the basic logic of this study is co-expression network analysis have advantages than transcriptome analysis and can further understand the data. However, the new findings or understandings about transcriptome data are not obvious in this manuscript.

Minor revisions:

1. In Page 3, the first paragraph, “Despite their tiny size of 246–401 nucleotides, ”. Now the size of the biggest viroid is 434 nt, which is much more than 401 nt.

Last paragraph, “Analyzing modular gene co-expression within these networks unveils the systemic functionality of genes [23, 24-29].” Why not [23-29]?

2. can’t find the reference [10].

3. “Emerging reports link viroid diseases to hormone pathways and transcription factor dynamics, disrupting plant gene expression landscapes [19] ”. The cited reference is not the most proper. Please check others.

4. Page 11, 3.2.1.1., “400 hundred genes” means 40 thousand genes?

5. Page 19, the second paragraph can be moved to results.

6. PLOS authors have the option to publish the peer review history of their article (what does this mean? ). If published, this will include your full peer review and any attached files.

**Do you want your identity to be public for this peer review?** For information about this choice, including consent withdrawal, please see our Privacy Policy .

Reviewer #1: No

Reviewer #2: No

---

## [Author Response · Author response to Decision Letter 1]

6 Nov 2024

Dear Editor and Reviewers,

We thank you for the thorough review and insightful comments on our manuscript “Dissecting the Role of bHLH Transcription Factors in the Potato Spindle Tuber Viroid-Tomato Pathosystem Using Network Approaches” submitted to PLOS ONE. We appreciate the opportunity to revise our work and have carefully considered each point raised during the review process. Below, we provide detailed responses to each comment from the reviewers and outline the revisions made to the manuscript to enhance its quality and clarity.

Reviewer #1: Comments and Responses

1. Comment: “Ten distinct gene co-expression modules, designated as M1 to M10, have varying sizes ranging from 216 to 23.” Please clarify if the number of genes is decreasing across the modules.

Response: We have clarified in the manuscript that the sizes of the modules do not follow a strictly decreasing order. The module sizes vary based on the number of genes they contain, which reflects the complexity and diversity of the co-expression patterns observed. This clarification has been added to the Results section “In our analysis of root tissue, we identified ten distinct gene co-expression modules, labeled M1 to M10, with sizes ranging from 23 to 216 genes”

2. Comment: “Module 2, comprising 123 genes, underscores the integral role of the bHLH transcriptional regulator, Solyc06g051550, in modulating iron uptake efficiency. This TF interacts with more than 400 genes in this module, highlighting its importance.” The phrase “400 hundred” is incorrect.

Response: The incorrect phrase “400 hundred” has been corrected to "more than 400 genes". We have thoroughly reviewed the document to ensure numerical accuracy.

3. Comment: “Two BHLH genes, Solyc06g051550 and SlbHLH036, act as central connectors.” Please ensure consistent use of the term “bHLH” for genes. Also, unify the naming of SlbHLH036 as Solyc05g006650.2 for consistency. The same is applied to bHLH083.

Response: We have revised the manuscript to consistently use the term “bHLH” for gene references. The naming of SlbHLH036 and bHLH083 gene symbols has been unified throughout the manuscript.

4. Comment: On Fig. 5A add a, b and c to determine interaction network and biological functions for M2, M4 and M5 respectively.

Response: We have updated Fig. 5A to include labels A, B, and C, corresponding to the interaction networks and biological functions of modules M2, M4, and M5, respectively.

5. Comment: In the text where you mention Fig.8 (A-D) replace the capital letters with lowcase as it is depicted in the figure.

Response: Response: The figure has been adjusted to use capital letters, consistent with the text where Fig. 8 (A-D) is referenced.

6. Comment: In the legend of Fig. 10 and inside the text replace capital letter in lowcase as it is depicted in the figure.

Response: The format for figures requires the use of capital letters. Therefore, the legend of Fig. 10 and all related textual references have been adjusted accordingly. Additionally, all figures have been edited to follow this style.

Reviewer #2: Major Revisions and Responses

1. Comment: In Page3, the second paragraph, “Yet, the presence of unknown genes in plant genomes poses challenges in data interpretation” It is difficult to understand the transition here. How does co-expression network analysis interpretate the data related with unknown genes? This study seems not integrate the data of any unknown genes but the known bHLH gene family. Additionally, from “This network forms a graph of…..” to the end, the introduction of co-expression network seems repetition and boring.

Response: We have clarified the Introduction section, emphasizing that our study focuses on known bHLH genes and that co-expression network analysis can help hypothesize the unknown functions of not completely annotated genes through their associations within the network. This transition has been revised to improve clarity.

“These high-throughput approaches have significantly advanced our understanding of the global effects of viroid infections, uncovering extensive alterations in gene expression across various plant systems [10]. However, the presence of genes with unknown functions in plant genomes continues to pose a challenge for data interpretation. In our study, we focused specifically on known bHLH genes, employing co-expression network analysis to mitigate this challenge. By clustering genes based on shared expression profiles, this method enables the prediction of their functions through associations within the network, providing a powerful tool for functional annotation [20-22]”

Comment: How does co-expression network analysis interpretate the data related with unknown genes?

Response: Co-expression network analysis is a powerful tool for interpreting data related to unknown genes, especially when their functions are not well characterized. This method relies on the assumption that genes with similar expression patterns across various conditions or samples are likely to be functionally related or co-regulated. Here is how co-expression network analysis can help interpret data related to genes with unknown functions:

Identification of functional modules. Co-expression network analysis groups genes into clusters or modules based on their expression patterns. Unknown genes that cluster with well-characterized genes can be inferred to have similar functions or be involved in the same biological processes.

Gene function prediction: By analyzing the co-expression patterns, unknown genes can be assigned potential functions. For example, if an unannotated gene is consistently co-expressed with genes involved in stress response, it may play a role in that process.

Network centrality measures: Co-expression networks use metrics like degree centrality, or betweenness centrality to identify key genes known as "hub" genes within a network. If an unannotated gene is a hub, it might be crucial in the regulation of the module, suggesting a significant role despite its unknown functional status.

Association with known pathways: Co-expression networks can be overlaid with pathway data. Unknown genes that are closely connected to genes in a specific pathway might be involved in that pathway, providing clues about their function.

“Co-expression network analysis groups genes into modules based on similar expression patterns, allowing the inference of functions for unknown genes clustered with well-characterized ones. By examining these patterns, potential functions for unannotated genes, such as involvement in stress response, can be predicted. Network centrality measures help identify "hub" genes, which may indicate key regulatory roles. Additionally, overlaying co-expression networks with pathway data can suggest unknown genes' involvement in specific biological pathways, providing further functional insights. Notably, genes with shared functions often exhibit robust correlations in their expression levels, laying the foundation for uncovering molecular pathways underlying diseases and conditions [31-33].”

2. Comment: In Page 4, the 4th paragraph, why only highlight “the tomato immune response” against viroid infection? Additionally, the sentence “we explore their involvement in host molecular mechanisms.” seems incomplete.

Response: We have broadened to highlight multiple pathways influenced by bHLH transcription factors, including immune responses as well as their roles in stress response, development, and signaling events during viroid infection.

“In this study, we aimed to identify co-expression modules of the bHLH transcription factor family, with a particular emphasis on their roles in diverse biological processes, including the tomato immune response to viroid infection. By positioning bHLH transcription factors (TFs) as hub genes, we explored their involvement in key host molecular mechanisms such as stress responses, development, and signaling pathways. Our primary objective was to uncover tissue- and strain-specific bHLH-guided transcriptional programs, highlighting both conserved and unique regulatory networks using network-based approaches”

Comment: Does bHLH play major roles in the resistance to viroid infection?

Response: Regarding the role of bHLH genes in viroid resistance, our study aimed to identify co-expression modules focusing on their functions in diverse biological processes. By positioning bHLH TFs as hub genes, we uncovered tissue- and strain-specific regulatory programs, emphasizing their roles in energy metabolism, membrane lipid repair in roots, and regulatory networks involved in leaf responses to PSTVd infection involved in the disruptions in cuticle function, shifts in metabolic processes, and alterations in biosynthesis pathways.

“Our results reveal that bHLH TFs are associated with regulation of genes involved in energy metabolism and membrane lipid repair functions in infected roots. Furthermore, our findings shed light on the intricate regulatory networks orchestrated by bHLH TFs in leaf tissue during PSTVd variant infections. The disruptions in cuticle function, shifts in metabolic processes, and alterations in biosynthesis pathways portray the plant's adaptive response to mitigate viroid impact”

Comment: “we explore their involvement in host molecular mechanisms.” seems incomplete.

Response: The sentence has been clarified to explain our exploration of these factors in host molecular mechanisms. “By positioning bHLH transcription factors (TFs) as hub genes, we explored their involvement in key host molecular mechanisms such as stress responses, development, and signaling pathways”

3. Comment: In Page 4, last paragraph, the first sentence, these is no solid molecular and genetic evidence that can prove that “bHLH TFs regulate photosynthesis and membrane lipid repair in infected roots”. The co-expression network analysis only shows association not causal relationship.

Response: Thank you for your insightful comment. We acknowledge that co-expression network analysis identifies associations rather than establishing causality. Our statement aimed to highlight the potential roles of bHLH TFs based on their co-expression with genes involved in photosynthesis and membrane lipid repair in infected roots. We have revised the text to clarify that these findings suggest a possible regulatory role of bHLH TFs, supported by association rather than direct molecular or genetic evidence. Future experimental studies, such as gene knockouts or overexpression analyses, will be necessary to experimentally confirm these regulatory functions.

“Our results reveal that bHLH TFs are associated with regulation of genes involved in energy metabolism and membrane lipid repair functions in infected roots”

4. Comment: Page 6, the first paragraph, why chose “Microarray datasets”? Most of the gene in microarray analysis should be known, while RNA-seq analysis can reveal some unknown genes.

Response: Thank you for your comment. While RNA-seq can identify novel genes, microarray data offer reliable insights, particularly for genes that are already annotated at the sequence level, as is the case with many genes in the tomato genome. Moreover, co-expression networks derived from microarray data enable functional inference for co-expressed genes, even when their annotations are incomplete, by leveraging their network associations. We selected the Corto algorithm, which requires a minimum of 20 samples to construct robust and reliable co-expression networks. This sample size criterion influenced our decision to use microarray datasets that met this requirement.

“Despite the advantages of high-throughput data, we selected these datasets because they represent the most extensive resource currently available for studying gene expression in tomato under viroid infection. Furthermore, the sample sizes (n) for each dataset are sufficiently robust to generate reliable coexpression networks, ensuring the identification of key regulatory nodes and gene modules involved in tissue-specific responses.”

Comment: Page 8, the second paragraph, the late sentence “This differential composition between the leaf and root……” This expression is too conclusive. No strong evidence supports “the tissue-specific regulatory roles played by bHLH TFs,”

Response: The statement has been revised to suggest potential tissue-specific roles of bHLH TFs, noting that further experimental validation is required.

“The observed differential composition between the leaf and root networks suggests that bHLH TFs may have tissue-specific regulatory roles, potentially influencing distinct physiological, developmental, and metabolic processes in a tissue-dependent manner. However, further experimental validation is required to confirm these roles and to elucidate the underlying mechanisms”

5. Comment: Page 9, the first paragraph, generally, there is no chloroplast in root, how to explain the finding that “Genes with a single regulator are implicated in Chloroplast, Photosystems I and II,”?

Response: We have clarified the reference to chloroplast-related genes and adjusted the text accordingly.

“The finding that genes with a single regulator are enriched in chloroplast functions, such as Photosystems I and II, indicates a specialized regulatory mechanism. Although chloroplasts are primarily found in photosynthetic tissues like leaves, these genes may also be active in non-photosynthetic tissues like roots under certain conditions. For example, roots may activate chloroplast-related pathways in response to stress, such as viroid infection, which reflects an evolutionary remnant of gene regulation or cross-talk between metabolic and stress pathways”

Discussion Section

“Indeed, chloroplast genes, generally active in leaves, have been observed in roots during stress conditions, such as shoot removal. Genes like GOX1 and GLDP1 are upregulated in roots in response to cytokinin signaling, suggesting they contribute to redox regulation and metabolic adjustments in non-photosynthetic tissues (Feng et al., 2021; Kobayashi & Masuda, 2013). This behavior represents an "evolutionary remnant of gene regulation," where chloroplast-related genes, once solely for photosynthesis, retain secondary functions and now aid plant stress responses across various tissues (Hayashi-Tsugane et al., 2014; Li et al., 2021)”

6. Comment: Page 11, the first sentence, “3.2.1 Modules that Exhibit Strong Symptom Induction in Response to the Severe Strain” What are the symptoms in root of tomato plants infected with viroid? and what associations of the following identified genes in this part with these symptoms?

Response: We have added descriptions of root symptoms, such as stunted growth and root necrosis, and discussed how identified genes may be associated with these symptoms.

“Three modules, M2, M4, and M5, showed significant regulatory shifts during the 24-day symptom development phase of the severe PSTVd strain, as illustrated in Figure 5(A). These modules displayed a coordinated surge in gene expression Figure 5(B) as root symptoms—such as stunted growth, root necrosis, and reduced root mass—worsened (24 dpi). These synchronized expression patterns could suggest that these genes may play a role in the plant's stress response and the progression of these symptoms”

7. Comment: Page14, “3.3. Exploring Leaf Tissue Gene Co-Expression Modules”. The symptoms induced by viroid infection in tomato plants are obvious in leaf. Why did not analyze the module associated with symptoms in leaf like the analysis of root?

Response: We conducted the same analysis for both root and leaf tissues. The approach involved identifying and exploring gene co-expression modules associated with observed symptoms in both tissues. By applying this consistent methodology, we ensured that the analysis of gene expression and the identification of key regulatory modules were directly comparable between roots and leaves, allowing us to gain understanding of the potential tissue-specific regulatory roles of the genes involved. We refer the reviewer to section “3.3. Exploring Leaf Tissue Gene Co-Expression Modules” where the analysis for the modules associated to leaf tissue are described.

8. Comment: Page 18, third paragraph, the sentence “This synchronization is an evolutionary

---

## [Decision Letter · Decision Letter 1]

22 Dec 2024

PONE-D-24-31516R1Dissecting the role of bHLH transcription factors in the potato spindle tuber viroid-tomato pathosystem using network approaches.PLOS ONE

Dear Dr. Hernandez-Rosales,

Thank you for submitting your manuscript to PLOS ONE. After careful consideration, we feel that it has merit but does not fully meet PLOS ONE’s publication criteria as it currently stands. Therefore, we invite you to submit a revised version of the manuscript that addresses the points raised during the review process.

We look forward to receiving your revised manuscript.

Kind regards,

Abozar Ghorbani, Ph.D

Academic Editor

PLOS ONE

Journal Requirements:

Reviewers' comments:

Reviewer's Responses to Questions

**Comments to the Author**

1. If the authors have adequately addressed your comments raised in a previous round of review and you feel that this manuscript is now acceptable for publication, you may indicate that here to bypass the “Comments to the Author” section, enter your conflict of interest statement in the “Confidential to Editor” section, and submit your "Accept" recommendation.

Reviewer #2: All comments have been addressed

Reviewer #3: (No Response)

2. Is the manuscript technically sound, and do the data support the conclusions?

Reviewer #2: Yes

Reviewer #3: Yes

3. Has the statistical analysis been performed appropriately and rigorously? 

Reviewer #2: Yes

Reviewer #3: Yes

4. Have the authors made all data underlying the findings in their manuscript fully available?

Reviewer #2: Yes

Reviewer #3: Yes

5. Is the manuscript presented in an intelligible fashion and written in standard English?

Reviewer #2: Yes

Reviewer #3: Yes

6. Review Comments to the Author

Reviewer #2: The revised manuscript has significantly improved and my questions have been resolved. After revise several minor mistakes about spelling, such as the loss of the left square bracket of references 11-15, it can be accepted.

Reviewer #3: I have received the article as review assignment, The article should be accepted after minor revisions

7. PLOS authors have the option to publish the peer review history of their article (what does this mean? ). If published, this will include your full peer review and any attached files.

**Do you want your identity to be public for this peer review?** For information about this choice, including consent withdrawal, please see our Privacy Policy .

Reviewer #2: **Yes: ** Zhang Zhixiang

Reviewer #3: No

---

## [Author Response · Author response to Decision Letter 2]

8 Jan 2025

Reviewer Report, PONE-D-24-31516R1

Thank you for assigning me the manuscript for review. It is a well-organized and a good piece of work that should be published, however I have few suggestions and few questions.

Response: Thank you very much for your valuable feedback and positive assessment of our manuscript. We appreciate your time and thoughtful suggestions, which we address below to further improve the manuscript.

Introduction Section:

1. Authors started with a phrase about viroids which is good but I suggest to add one line for their lack of protein expression and characteristics like this. Very briefly.

Response: Thank you for the suggestion. We have incorporated a brief sentence highlighting the lack of protein expression in viroids and their reliance on host enzymatic machinery as follows:

“This lack of protein expression makes viroids unique among plant pathogens, relying entirely on host enzymatic machinery for replication and systemic movement”

2. how widespread is PSTVd in major tomato-growing regions? Please mention this detail

Response: Thank you for this observation. We have added the following paragraph in order to detail the widespread presence of PSTVd in major tomato-growing regions, emphasizing its global distribution and the associated risks due to international trade and insufficient phytosanitary controls as follows:

“PSTVd has been reported in major tomato-growing regions worldwide, including Europe, North and South America, Asia, and Africa, where it poses a significant threat to agricultural productivity (https://gd.eppo.int/taxon/PSTVD0/distribution; accessed on January 2025). Outbreaks in these regions are often associated with the international trade of infected plant material and inadequate phytosanitary controls, highlighting the importance of global surveillance and stringent management practices.

3. why were tomato chosen as the model system for this study?

Response: Tomato (Solanum lycopersicum) was selected as the model system for this study due to its economic importance as a staple crop worldwide and its well-characterized genetic and transcriptomic resources. Additionally, tomato is a natural host of PSTVd, making it highly relevant for studying viroid-host interactions. The availability of robust genomic tools and its susceptibility to PSTVd infection provide an ideal framework for investigating the molecular mechanisms of viroid pathogenesis and its impact on plant physiology.

We have rephrase this paragraph in the introduction as follows:

“An illustrative instance can be observed in tomato (Solanum lycopersicum), a vital global agricultural commodity that yielded more than 130,812,947 million dollars in 2022 (http://faostat.fao.org/, accessed in May 2024). This staple crop faces a tangible threat from viroids, underscoring the potential impact of these infectious agents on essential food production systems.Tomato was selected as the model system for this study due to its economic importance as a staple crop worldwide and its well-characterized genetic and transcriptomic resources. This choice is particularly relevant given that Potato spindle tuber viroid (PSTVd), one of the most studied viroids, has a significant economic impact on solanaceous crops, including tomato.”

Materials and Methods:

1. The mention of tools like Corto, CEMiTool, and Mfinder is clear. However, briefly explaining why these specific tools were chosen over alternatives could strengthen the methodology's justification.

Response: For the analysis of transcriptional regulatory networks, Corto was chosen over ARACNe due to its ability to accurately reconstruct networks with smaller sample sizes, a critical consideration given the dataset used in this study. To identify co-expression modules, CEMiTool was selected instead of WGCNA, as it provides an automated pipeline that simplifies the identification, visualization, and functional annotation of co-expression modules. For network motif analysis, Mfinder was employed for its specific capability to detect and analyze network motifs, offering deeper insights into key regulatory patterns, which are less accessible with general-purpose network analysis tools. To strengthen the justification for our methodological choices, we clarified the selection of Corto, CEMiTool, and MFinder as follows:

“For the network analyses, we used Corto, CEMiTool, and MFinder due to their specialized capabilities. Corto was used to reconstruct tissue-specific transcriptional regulatory networks, emphasizing bHLH interactions (Steps (A)–(B)), leveraging prior biological knowledge for accurate inference. CEMiTool identified and enriched co-expression modules, providing system-wide insights into gene interactions (Step (C)). Bipartite networks for each tissue were then constructed using Python scripts derived from CEMiTool output files (Step (D)). Finally, MFinder detected network motifs within the co-expression networks, uncovering key regulatory patterns (Step (E)). These tools were chosen for their precision, ease of implementation, and alignment with the study's objectives.”

2. Mention the threshold used for Spearman correlation coefficients and explain how it was determined (e.g., significance level, biological relevance).

Response: Thank you for the suggestion; we have revised and rewritten this section as follows:

“The Corto algorithm, available on the CRAN repository of R packages, was used to infer gene regulatory networks (GRNs) from tissue-specific expression matrices [40]. For this analysis, we used an integrated expression matrix and the PlantTFDB tomato transcription factor list. Corto employs Spearman correlation to identify gene associations and applies the Data Processing Inequality (DPI) to eliminate indirect interactions, with bootstrapping to ensure robustness. The leaf network included 27 samples and 8080 features, while the root network had 26 samples and 8080 features. Spearman correlation thresholds of 0.837 (leaves) and 0.828 (roots) were applied, alongside a p-value threshold of 1 × 10⁻⁸ was applied to filter significant edges, and 100 bootstraps were performed to ensure robustness. From these global GRNs, subnetworks focusing on bHLH-TFs and their interactors were extracted, enabling detailed analysis of tissue-specific regulatory dynamics under PSTVd-induced stress”

3. The use of Gprofiler is appropriate, but details on how significance thresholds (e.g., FDR values) were determined would enhance reproducibility.

Response:We have incorporated the following sentence:

“…using an FDR threshold of < 0.05 to ensure statistically significant results…”

4. Were time points (17, 24, 49 dpi) treated independently, or were longitudinal patterns analyzed?

Response: Thank you for the question; we have clarified how time points were handled in each network type as follows:

“Time points (17, 24, and 49 dpi) were analyzed differently depending on the network type. In the global CEMiTool network, data from all-time points were integrated to create a comprehensive co-expression network. For module-specific networks, time points were treated separately to capture dynamic changes across stages, resulting in dynamic networks. Finally, in the bipartite networks, data from all-time points were combined again to provide an integrated overview of significant interactions”

Discussion section

1. how does the activity of bHLH TFs in roots contribute to stress adaptation versus their role in leaves in energy homeostasis or defense? Providing more explicit comparisons could clarify the versatility of these TFs in managing stress in different plant organs.

Response: Thank you for your insightful comments. We have expanded the discussion to provide a more explicit comparison of bHLH TF activity in roots and leaves. Specifically, we now highlight how bHLH TFs in roots contribute to stress adaptation by activating chloroplast-associated pathways for redox homeostasis and metabolic stability, while in leaves, these TFs prioritize energy reallocation and defense mechanisms, such as photosynthesis regulation and secondary metabolite biosynthesis. To address this, we inserted the following section:

The distinct activity of bHLH TFs in roots and leaves during PSTVd infection underscores their versatility in managing tissue-specific stress responses. Bipartite network analysis in this study reveals complementary regulatory dynamics in these organs, highlighting their specialized roles in stress adaptation.

For instance in roots, GO analysis reveals the activation of chloroplast-associated pathways, including Photosystems I and II, despite the non-photosynthetic nature of this tissue. As previously discussed, this activation likely represents an evolutionary remnant of gene regulation or a form of metabolic crosstalk, allowing roots to leverage chloroplast-related pathways to maintain redox homeostasis and redirect energy metabolism toward survival mechanisms. Key regulators such as SlbHLH052 Fer involved in heme binding and iron homeostasis, illustrate the critical importance of structural resilience and metabolic adaptation in root stress responses to viroid infection.

In contrast, leaves exhibit a more complex regulatory network, involving 30 bHLH TFs and 1970 target genes. A unique feature of the leaf network is the presence of genes regulated by more than four bHLH TFs, highlighting the intricate coordination required to balance energy homeostasis and defense mechanisms. Leaf-specific bHLH TFs, such as GBOF-1 (Solyc06g072520.1) and SlbHLH022 (Solyc03g097820.1) prioritize metabolic regulation, and secondary metabolite biosynthesis to facilitate rapid responses to biotic stress.

The modular co-expression analysis reveals distinct tissue-specific dynamics in the plant’s response to viroid infection, with root and leaf hub genes exhibiting entirely unique profiles. This specialization underscores the plant’s systemic strategy to deploy tailored molecular mechanisms to address the specific environmental and metabolic challenges faced by each organ.

In roots, bHLH TFs integrate with jasmonic acid (JA) and ethylene signaling pathways to regulate nutrient uptake and maintain structural stability, ensuring effective stress adaptation. For example, Solyc06g034370.1 hub gene (Pectinesterase) modifies the cell wall, providing enhanced mechanical support and increasing root resilience during viroid infection. Additionally, ethylene-responsive genes, such as Solyc09g075420.2 (Ethylene-responsive transcription factor 2b), play a pivotal role in integrating ethylene signaling to further support redox balance and structural integrity.

In contrast, leaf-specific hub genes, such as Solyc06g073260.2 (NAD-dependent epimerase) and Solyc10g078300.1 (Single-stranded nucleic acid binding R3H protein), prioritize immune signaling and secondary metabolite biosynthesis. These processes effectively reallocate metabolic resources to strengthen pathogen resistance, aligning with salicylic acid (SA)-dependent systemic acquired resistance (SAR) and enhancing both localized and systemic defense mechanisms. The interplay of bHLH TFs with key hormonal pathways, including JA, ethylene, auxin and, SA, refines tissue-specific stress responses by coordinating distinct molecular and physiological processes.

In roots, modules associated with severe PSTVd infection at 24 dpi highlight key stress-adaptive pathways, including ubiquitin-mediated defense, and cell wall dynamics. These modules display temporal dynamics, with early induction and subsequent repression during recovery, particularly in genes related to photosynthesis, such as photosystem I reaction center subunit V (Solyc07g066150) and chlorophyll a-b binding protein 7 (Solyc10g006230). In leaves, most modules show repression, except for Module 2 (M2), which is linked to defense responses. Modules M1 and M3 Emphasize repression of chloroplast biogenesis and metabolic regulation, respectively, while Module 5 (M5), regulated by bHLH083 (Solyc03g118310), underscores the role of carbohydrate metabolism and energy homeostasis. Repression of chloroplast-related genes, including those involved in chloroplast biogenesis, reflects a strategic reallocation of resources from photosynthetic processes to defense, further emphasizing the prioritization of pathogen resistance in this tissue.

The differential regulation of chloroplast-related genes in roots and leaves highlights their distinct functional priorities during stress. In roots, the activation of chloroplast-associated pathways facilitates metabolic adjustments essential for stress tolerance, enabling redox homeostasis and energy redirection to survival mechanisms. In contrast, leaves repress these pathways to efficiently reallocate energy and resources toward defense, prioritizing the production of secondary metabolites and activation of immune responses. This divergence exemplifies a coordinated systemic response to viroid infection, with roots focusing on maintaining metabolic stability, while leaves concentrate on pathogen resistance and energy conservation. Moreover, these processes, including chloroplast biogenesis and repair, not only sustain photosynthetic efficiency under stress but also drive the biosynthesis of secondary metabolites such as flavonoids and alkaloids, which are crucial for pathogen defense and overall plant resilience.

2. It is suggested to insert a sentence,

"The GO analysis identified several key categories, such as stress response, energy metabolism, and protein modification, which are significantly enriched in the identified gene modules, suggesting a broad functional role in managing PSTVd-induced stress."

Response: Thank you for the suggestion. We have adapted and incorporated the proposed sentence as follows:

"The GO analysis further supports this, identifying key functional categories such as stress response, energy metabolism, and protein modification, which are significantly enriched in the identified gene modules. This enrichment underscores the broad functional role of these modules in orchestrating the plant's systemic response to PSTVd-induced stress, integrating regulatory signals to optimize both defense and survival strategies."

3. Please discuss the chloroplast genes, as an evolutionary development

Response: Thank you for your suggestion. In the discussion, we have addressed the evolutionary significance of chloroplast gene regulation as follows:

“For instance, in roots, GO analysis reveals the activation of chloroplast-associated pathways, including Photosystems I and II, despite the non-photosynthetic nature of this tissue. This activation likely represents an evolutionary remnant of gene regulation or a form of metabolic crosstalk, allowing roots to leverage chloroplast-related pathways to maintain redox homeostasis and redirect energy metabolism toward survival mechanisms. Key regulators, such as SlbHLH052 Fer, involved in heme binding and iron homeostasis, illustrate the critical importance of structural resilience and metabolic adaptation in root stress responses to viroid infection.”

Additionally, we highlighted the contrasting role of chloroplast genes in leaves:

“Repression of chloroplast-related genes, including those involved in chloroplast biogenesis, reflects a strategic reallocation of resources from photosynthetic processes to defense, further emphasizing the prioritization of pathogen resistance in this tissue. This divergence exemplifies a coordinated systemic response to viroid infection, with roots focusing on maintaining metabolic stability, while leaves concentrate on pathogen resistance and energy conservation.”

These sections illustrate how chloroplast-related genes, originally dedicated to photosynthesis, may have evolved secondary roles in stress adaptation across different tissues, reflecting an evolutionary and functional specialization in response to biotic stress.

4. While discussing the role of bHLH TFs, it might be useful to briefly mention how they interact with other transcription factors or signalling pathways involved in plant defence.

Response: Thank you for your suggestion. We have incorporated the following discussion on the interactions between bHLH TFs, targets and other transcription factors and signalin

---

## [Decision Letter · Decision Letter 2]

20 Jan 2025

Dissecting the role of bHLH transcription factors in the potato spindle tuber viroid-tomato pathosystem using network approaches.

PONE-D-24-31516R2

Dear Dr. Hernandez-Rosales,

We’re pleased to inform you that your manuscript has been judged scientifically suitable for publication and will be formally accepted for publication once it meets all outstanding technical requirements.

Kind regards,

Abozar Ghorbani, Ph.D

Academic Editor

PLOS ONE

Additional Editor Comments (optional):

Reviewers' comments:

Reviewer's Responses to Questions

**Comments to the Author**

1. If the authors have adequately addressed your comments raised in a previous round of review and you feel that this manuscript is now acceptable for publication, you may indicate that here to bypass the “Comments to the Author” section, enter your conflict of interest statement in the “Confidential to Editor” section, and submit your "Accept" recommendation.

Reviewer #3: All comments have been addressed

2. Is the manuscript technically sound, and do the data support the conclusions?

Reviewer #3: Yes

3. Has the statistical analysis been performed appropriately and rigorously? 

Reviewer #3: Yes

4. Have the authors made all data underlying the findings in their manuscript fully available?

Reviewer #3: Yes

5. Is the manuscript presented in an intelligible fashion and written in standard English?

Reviewer #3: Yes

6. Review Comments to the Author

Reviewer #3: i have already presented that comments for the reviewers in first round .The comments have been addressed, it is recommended for acceptance

7. PLOS authors have the option to publish the peer review history of their article (what does this mean? ). If published, this will include your full peer review and any attached files.

**Do you want your identity to be public for this peer review?** For information about this choice, including consent withdrawal, please see our Privacy Policy .

Reviewer #3: **Yes: ** Muhammad Taimoor Shakeel

---

## [Editor Report · Acceptance letter]

PONE-D-24-31516R2

PLOS ONE

Dear Dr. Hernandez-Rosales,

I'm pleased to inform you that your manuscript has been deemed suitable for publication in PLOS ONE. Congratulations! Your manuscript is now being handed over to our production team.

Kind regards,

on behalf of

Dr. Abozar Ghorbani

Academic Editor

PLOS ONE